# In situ captured antibacterial action of membrane-incising peptide lamellae

Kamal el Battioui[1,2], Sohini Chakraborty[1], András Wacha [1], Dániel Molnár[3,4], Mayra Quemé-Peña[1,2], Imola Cs. Szigyártó[1], Csenge Lilla Szabó [2,5], Andrea Bodor [5], Kata Horváti[6], Gergő Gyulai [6,7], Szilvia Bősze[8], Judith Mihály[1], Bálint Jezsó [1,9], Loránd Románszki [1], Judit Tóth[3,10], Zoltán Varga[1,11], István Mándity[1,12], Tünde Juhász[1] & Tamás Beke-Somfai [1] ✉

Developing unique mechanisms of action are essential to combat the growing issue of antimicrobial resistance. Supramolecular assemblies combining the improved biostability of non-natural compounds with the complex membrane-attacking mechanisms of natural peptides are promising alternatives to conventional antibiotics. However, for such compounds the direct visual insight on antibacterial action is still lacking. Here we employ a design strategy focusing on an inducible assembly mechanism and utilized electron microscopy (EM) to follow the formation of supramolecular structures of lysine-rich heterochiral $\beta^3$-peptides, termed lamellin-2K and lamellin-3K, triggered by bacterial cell surface lipopolysaccharides. Combined molecular dynamics simulations, EM and bacterial assays confirmed that the phosphate-induced conformational change on these lamellins led to the formation of striped lamellae capable of incising the cell envelope of Gram-negative bacteria thereby exerting antibacterial activity. Our findings also provide a mechanistic link for membrane-targeting agents depicting the antibiotic mechanism derived from the in-situ formation of active supramolecules.

With the overuse of conventional antibiotics, there is a rapid emergence of pathogens immune to standard treatment strategies[1,2]. The lengthy and expensive development of related drugs is much slower than the spread of antimicrobial resistance (AMR), making this phenomenon one of the major health concerns of World Health Organization (WHO)[3,4]. Ideally, compounds with complex killing mechanisms akin to those of host antimicrobials[5,6] against which the development of AMR is heavily hindered, should be synthesized[7,8]. Most of these natural antibiotics[9] are typically amphiphilic and have a peptidic backbone with rotatable bonds that can exhibit a conformational change upon binding to bacterial membranes. This change in secondary structure could also be exploited to form assembled

[1]Institute of Materials and Environmental Chemistry, HUN-REN Research Centre for Natural Sciences, Budapest H-1117, Hungary. [2]Hevesy György Ph.D. School of Chemistry, Eötvös Loránd University, Budapest H-1117, Hungary. [3]Institute of Molecular Life Sciences, HUN-REN Research Centre for Natural Sciences, Budapest H-1117, Hungary. [4]Doctoral School of Biology and Institute of Biology, Eötvös Loránd University, Budapest H-1117, Hungary. [5]ELTE Eötvös Loránd University, Institute of Chemistry, Analytical and BioNMR Laboratory, Budapest H-1117, Hungary. [6]MTA-HUN-REN TTK "Momentum" Peptide-Based Vaccines Research Group, Institute of Materials and Environmental Chemistry, Research Centre for Natural Sciences, Budapest H-1117, Hungary. [7]ELTE Eötvös Loránd University, Institute of Chemistry, Laboratory of Interfaces and Nanostructures, Budapest H-1117, Hungary. [8]HUN-REN ELTE Research Group of Peptide Chemistry, Hungarian Research Network, Eötvös Loránd University, Budapest, Hungary. [9]ELTE-MTA "Momentum" Motor Enzymology Research Group, Department of Biochemistry, Eötvös Loránd University, Budapest, Hungary. [10]Department of Applied Biotechnology and Food Sciences, Budapest University of Technology and Economics, Budapest H-1111, Hungary. [11]Department of Physical Chemistry and Materials Science, Budapest University of Technology and Economics, Műegyetem rkp. 3, Budapest 1111, Hungary. [12]Department of Organic Chemistry, Faculty of Pharmacy, Semmelweis University, Budapest H-1092, Hungary. ✉e-mail: beke-somfai.tamas@ttk.hu

constructs with diverse overall effects beyond that of a monomer[10,11]. A drawback of this natural peptide backbone, however, is a resulting low biostability, limiting widespread therapeutic use[12,13]. Thus, related non-natural compounds have been produced which can withstand proteolytic degradation[14–16]. β-peptides are the closest homologues to natural α-peptides with high efficacy for assembly formation[17,18], making them ideal for development of in situ forming bactericidal assemblies. To progress towards such antimicrobial supramolecules, we aim to reach a building unit that could assemble into the active morphology on membranes of some clinically challenging strains, such as members of the Gram-negative *Enterobacteriaceae*[19]. Gram-negative species generally exhibit greater resistance to antibiotics as they contain an extra outer membrane comprised of lipopolysaccharides (LPS)[20]. LPS act as a barrier to shield bacterial cells from antibiotics and they are anchored to the membrane by their lipid A region, which is rich in negatively charged phosphate moieties[21]. Note, that owing to their fluidity and diverse composition, lipid bilayers are particularly challenging to target[22], and this is why we focused on the lipid A motif as it may prove to be an ideal target for cationic agents. In order to exploit the interphosphate distances on lipid A[23], we develop lysine-rich heterochiral β³-peptides that could utilize the phosphate groups on LPS for in situ coassembly formation.

## Results

### Design strategy and initial theoretical assessment

We have recently reported a series of water soluble, membrane active acyclic β³-peptides with alternating chirality, which could assemble into oligomers[24]. The individual peptides adopted a zig-zag backbone conformation stabilized by an intramolecular salt bridge between a lysine and a glutamate side-chain. This zig-zag secondary structure, with residues on the two sides of the backbone, is similar to the β-strand conformation of natural peptides, which makes it suitable for forming intermolecular hydrogen bonds between neighboring peptides akin to amyloid fibrils[25]. Here, we aimed to improve this basic scaffold and initiate assembly formation assisted by external phosphate groups by making the starting sequence cationic. We tested the hypothesis of phosphate-assisted coassembly formation first with molecular dynamic (MD) simulations on β³-lysine rich hexapeptide and octapeptide analogues (Fig. 1a), named lamellin-2K and lamellin-3K (or 2K and 3K, respectively in short), as a single phosphate ion is able to coordinate multiple lysines and other cationic residues as in e.g., RecA-like helicases[26,27]. By having several lysine residues in close proximity to each other, we attempted to reach a molecule that is initially in random conformation but could become ordered upon interactions with phosphates. Simulations in water showed preference for random coil conformation for both 2K and 3K due to repulsion of the positively charged side chains (Fig. 1b, c). In further simulations with the presence of phosphates, the relative position of the lysine side chain $NH_3^+$ groups become strongly localized where two neighboring amino groups coordinated a single phosphate (Supplementary Fig. 1a). As a following step, we tested oligomerization on eight 3K monomers and multiple phosphate ions (Fig. 1c, Supplementary Fig. 1b and Supplementary Table 1). The simulations showed that multiple binding to phosphates enabled a zig-zag conformation, which could eventually lead to the formation of a 3K-phosphate coassembly, similar to the case where Glu and Lys residues could form an intramolecular salt bridge without the requirement of external coordinating ions[24].

### Structure and assembly in various environments

Experimental characterization of the structural differences for the designed peptides was initially tested in several aqueous environments, including pure water, NaCl solution, phosphate solution, and phosphate buffered saline (PBS). We first tested the relative spectral differences by circular dichroism spectroscopy (CD), as this can immediately report on major changes in conformation[28,29]. Both of the

peptides pointed towards the predominance of a random coil structure in phosphate-free conditions while they turned into a more ordered structure in phosphate rich media. Particularly, the CD signature of lamellin-3K shows a significantly increased intensity at ~207 nm that occurs in a similar manner as e.g., upon formation of β-sheet rich amyloid assemblies for natural peptides (Supplementary Fig. 2a, b)[30,31]. This effect was also followed sequentially through several concentration dependence measurements on 3K-PBS (Supplementary Figs. 3, 4). These conformational changes were also assessed using infrared spectroscopy (IR). The spectral pattern with a main amide I band component centred at ~1650 cm⁻¹ is compatible with dominant random coil or strand-like structures, while a shoulder band at ~1680 cm⁻¹ can be assigned to turn motives similarly to natural peptides[32] or to backbone amide C = O groups not involved as acceptors in H-bonding (Supplementary Fig. 5)[33,34]. In the presence of phosphates, the relative intensity of the latter is significantly reduced for 3K. Further on, a low wavenumber component indicative of strong intermolecular H-bonding appears for both peptides, being more intense for 2K in phosphates. It seems plausible that the phosphate moieties facilitate formation of a more ordered structure at the expense of other conformations, which is in line with the above MD and CD results and also similar to our previous results where sheet-like oligomers built up from β-peptides in zig-zag conformation[24]. The same major peaks with very similar relative intensity ratios were also observed for sheet forming cyclic β-peptides when their sequence was elongated from trimers into pentamers and heptamers[33]. Notably, helical β-peptides display rather distinct band ratios for these, where the ~1650 cm⁻¹ component is less dominant[34]. Furthermore, the CD signature for 2K-PBS is very similar to that observed by Martinek et al. for a cyclic sheet-rich β-peptide forming long layers, supporting a non-helical, strand-like conformation (Supplementary Fig. 2a)[35].

NMR spectroscopic data collected in pure water (Supplementary Fig. 6) also support the random coil conformation of both lamellin-2K and 3K. Low signal dispersion for both ¹H (Supplementary Fig. 6a–c) and ¹³C is a characteristic property of disordered peptides and proteins. Pulsed-field gradient NMR (PFG-NMR) derived diffusion coefficients (Supplementary Table 2) correspond to the *D* values of intrinsically disordered proteins (IDPs) with the respective molar masses under similar experimental conditions[36,37]. The ROESY spectra showed only sequential crosspeaks, between the $H^\alpha$ and $H^\beta$ of the $i^{th}$ residue and the amide ($H^N$) of the $(i+1)^{th}$ residue (Supplementary Fig. 6d, e), but lacked $H^N$-$H^N$ crosspeaks and there was no sign of long-range $(i+2, i+3)$ interactions either, that would normally be present in a well-defined 3D fold (Supplementary Fig. 6f, g)[38,39]. The small chemical shift difference between the diastereotopic $H^\alpha$ atoms, and the medium $^3J_{H\beta\text{-}H\alpha}$ couplings indicate a free rotation around the $C^\alpha$-$C^\beta$ bond, supporting the disordered nature of the peptide[40]. The only rigidity detected by NMR is along the $C^\beta$-N bond, the large values of $^3J_{HN\text{-}H\beta}$ couplings (9.3–9.7 Hz) suggest a nearly anti-periplanar arrangement of $H^N$ and $H^\beta$ protons (Supplementary Table 3)[34]. Adding phosphate to the media reduced the solubility of both compounds, the maximal concentration of 2K was ~250 μM, while the solubility of 3K fell to ~20 μM based on the leucine methyl signals (Supplementary Fig. 7e). The signal dispersion of the 1D ¹H spectrum remained low with minor chemical shift perturbance compared to pure water, while $^3J_{HN\text{-}H\beta}$ and $^3J_{H\beta\text{-}H\alpha}$ scalar couplings did not suffer a significant change. Diffusion coefficients do not report on a significant compaction, so the soluble lamellins remained in an extended conformation. 2D homonuclear NMR characterization in PBS was feasible for only 2K, and the acquired ROESY spectrum was very similar to that in pure water (Supplementary Fig. 7), supporting an extended conformation for the soluble part of 2K.

To test whether phosphate could induce an assembly formation, for both 3K and 2K we used fluorescence spectroscopy by evaluating the binding capacity of the anionic hydrophobic probe 1-anilinonaphthalene-8-sulfonate (ANS)[41]. A higher fluorescence intensity

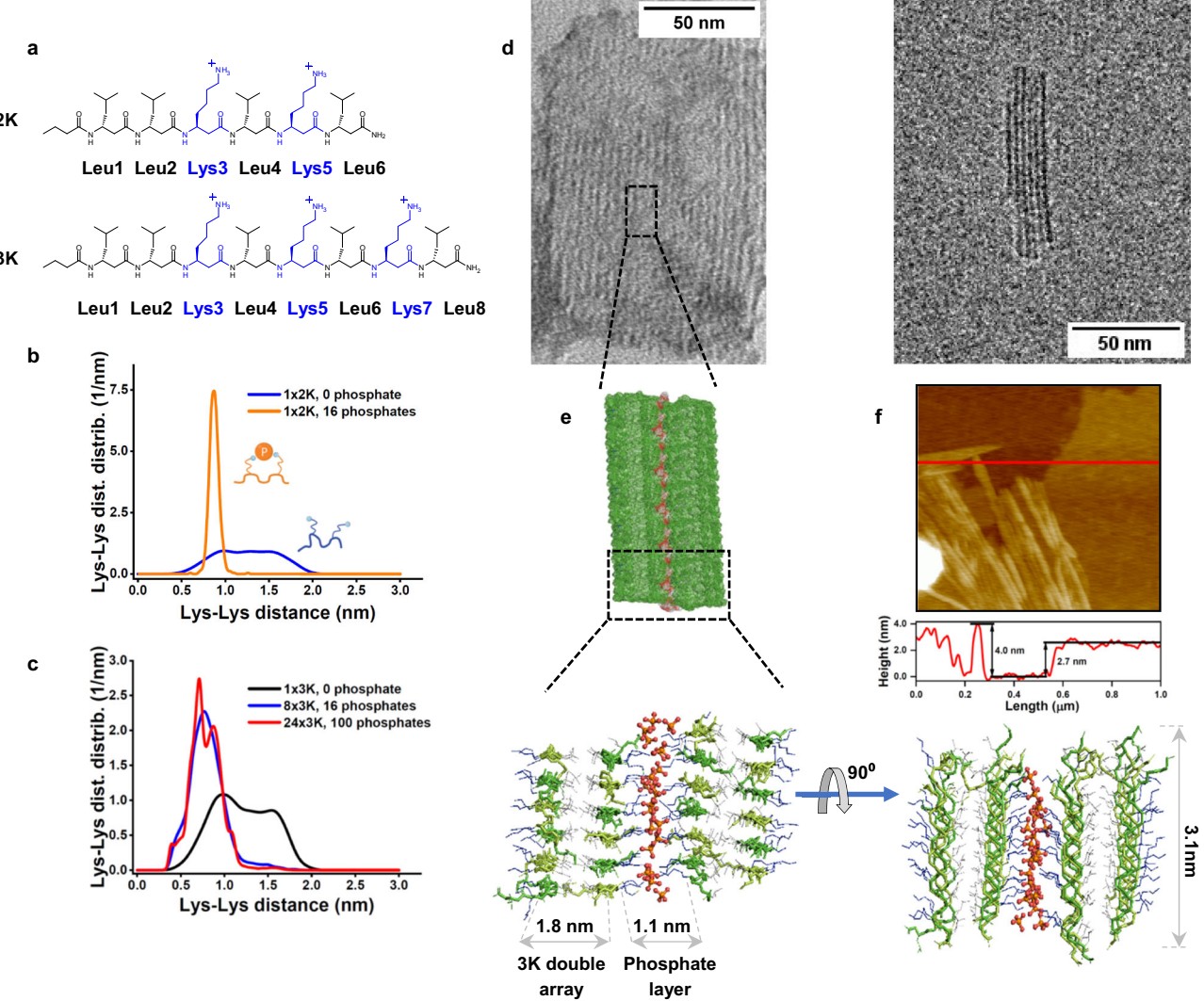

**Fig. 1 | Structure and morphology of the lamellin-2K and lamellin-3K assemblies.** **a** Sequence of 2K and 3K with the leucines marked in black and the lysines marked in blue. **b** Distance distribution of lysine amine nitrogen atoms of 2K detected in MD simulations in the absence and presence of phosphate ions displayed as blue and orange lines respectively; inset shows schematic representation of the backbone conformations, the orange sphere stands for the phosphate ion and the small blue spheres indicate the ε-amino group of lysines. **c** Distance distribution of lysine amine nitrogen atoms of one 3K molecule in the absence of phosphate ions displayed as a black line, eight 3K molecules in the presence of 16 phosphate ions displayed as a blue line and twenty four 3K molecules in the presence of 100 phosphate ions displayed as a red line as detected in MD simulations. For 3K, Lys3-Lys5 and Lys5-Lys7 distances were averaged. **d** Representative negative-stain TEM (NS-TEM) (left. Scale bar: 50 nm)

and cryo-EM (right. Scale bar: 50 nm) images of the lamellar morphologies formed by 3K (125 μM) in phosphate solution. Measurements were repeated three times and similar results were obtained. **e** Snapshot of MD simulation on the 3K-phosphate coassembly (24 x 3K + 100 phosphates). TOP: Schematic mesh representation of the 3K-phosphate (or 3K-PBS) arrays based on electron density of the assemblies in the MD simulation. BOTTOM: side (left) and top (right) view of the arrays, with main dimensions displayed. Peptide backbones are displayed as green sticks, lysine and leucine side chains as blue and grey lines, respectively. Phosphate ions are ball and sticks, where oxygen atoms are red, and phosphorus atoms are orange. **f** AFM image of 3K (12.5 μM) in PBS, and a representative cross-section height profile along the marked line. Measurements were repeated three times and similar results were obtained.

was observed for the media with phosphate content indicating effective binding of the probe to the assemblies likely providing a hydrophobic binding site for the apolar moiety of the dye (Supplementary Fig. 8). Together with the above measurements, it is suggested that these β-peptide systems are initially in random coil monomeric states, but upon addition of phosphate ions to the solution (i.e. solely phosphates or as PBS buffer), this arrives to a more ordered stage. Since rapid assembly formation prevented atomic insight by NMR, and considering that CD and IR spectra could sometimes be very similar for various β-peptide secondary structures[38,39,42,43], we have performed additional MD simulations and experiments to gain further structural details. First, we tested extensively the helix forming ability of 3K, using our recently developed force field that could predict the correct fold for various β-peptide sequences[44–46]. These simulations all strongly indicated that the inherent

properties of 3K make helical conformations unfavorable (Supplementary Figs. 9–11). To address formation of larger sheet-like assemblies, tetracosameric (24-mer) models were built based on the initial octameric assemblies. The simulations showed stable formation of lamellin-3K double arrays in parallel orientation. The arrays were stabilized by H-bonds along the long axis of the assembly, and also by a hydrophobic core of leucine side chains (Fig. 1c, e). These double arrays are connected by a layer of phosphate ions that coordinate the lysines. The accumulation of phosphate ions on the outer edges of the assembled peptide arrays suggests that in principle this oligomeric motif is repeated in the coassembly. The zig-zag lamellin-3K molecules span ~3.1 nm, whereas the width of the double arrays and the hydrophilic lysine-phosphate regions are ~1.8 nm and ~1.1 nm, respectively (Fig. 1e). Interestingly, while similar conformations were predicted by theoretical

studies for sheets by Seebach et al.[43] and by us[47,48], however, here we only see a parallel orientation and the torsional angles are not the same (Supplementary Fig. 11). This structure lies closest to the parallel sheet orientation suggested for cyclic β-peptides with amino acids having alternating chirality[33].

The above MD results strongly suggest that phosphates can drive 3K molecules into non-helical conformations and induce sheet-like ordered assemblies. To further strengthen lack of helical structures, and to also explore whether biological coordinating ions other than phosphate could induce similar lamellae formation, we tested 3K with a heparin mimic molecule, suramin[49–51]. It is known that sulfated glucosaminoglycans can induce helical conformation for short antimicrobial peptides[52] and suramin is one of the strongest helix-inducing compound for these[53–55]. The resulting CD spectra are close to 3K in water and in NaCl, thus this heparin mimic could not induce a helical 3K assembly, and this system is distinct from 3K-PBS (Supplementary Fig. 12).

## Morphology and molecular packing

Based on the structural information above, we addressed the morphology of the coassemblies formed in phosphate solution. Electron micrographs of both peptides in PBS displayed striped lamellae (Fig. 1, Supplementary Figs. 13, 14). This is in strong contrast to the images obtained in phosphate-free media, which lacked ordered morphologies (Supplementary Fig. 15a, b). For lamellin-3K in phosphate, there is occurrence of similar striped lamellar morphology which forms semi-ordered aggregates (Supplementary Fig. 15). The development of these well-defined rectangular morphologies was more pronounced for lamellin-3K (Supplementary Fig. 14c, d), whereas lamellin-2K in PBS formed elongated striped lamellae, though the striped packing morphology are very similar for these (Supplementary Fig. 13). The pattern seen in negative staining (NS), is anticipated to originate from the alternating chirality of the repetitive β-(3$R$)-Leu-β-(3$S$)-Lys motifs[24], which enables a phosphate coassembly formation. The bright stripes result from ordered, hydrophobic peptide regions[56], with an average width of $1.60 \pm 0.22$ nm (Supplementary Fig. 16 and Supplementary Table 4). The darker ones, with a width of $1.13 \pm 0.13$ nm, could be assigned to hydrophilic layers of arranged phosphates. To exclude the effect of the staining complex, alternative staining with phosphotungstic acid (PTA) was also carried out, which resulted in the same striped lamellar morphology (Supplementary Fig. 14d). Furthermore, 2K and 3K-phosphate coassemblies were also addressed using a cryo-EM setup, to acquire images in an as native as possible environment. The cryo-EM images displayed similar morphologies as for NS-TEM, both for 2K and for 3K, however for the latter the rectangular lamellae could reach several hundred nanometers in length (Supplementary Fig. 13d, e and Fig. 17). The width of the repetitive bright and dark arrays was found to be ~2.7 nm, from the EM images. To test whether interaction with suramin could also result in similar lamellae, NS-TEM images were also recorded for 3K-suramin. These demonstrate completely different morphologies confirming the related CD results and that the 3K-suramin is distinct from the 3K-PBS morphologies (Supplementary Fig. 12d). 3K both alone and in PBS solution was also investigated by small angle X-ray scattering (SAXS). The obtained curves indicated a regular periodic morphology, with a repeat distance of 2.86 nm using Bragg's equation, which lies close to the EM values. The comparison of the solution phase 3K alone and 3K-PBS also demonstrated that the latter is in a more ordered state (Supplementary Fig. 18 and Supplementary Table 5). To obtain information on the thickness of the lamellae, AFM topography was performed at multiple lamellin-3K concentrations. A 2.5−3.0 nm thick monolayer could be observed for all employed setups, where the higher concentrations contained increasingly more packed layers (Fig. 1f and Supplementary Fig. 19). To confirm the presence of similar morphologies in solution, liquid AFM measurements were also performed, which demonstrate that the dried and solution phase lamellar morphologies are comparable at the investigated concentration (Supplementary Fig. 20). Note that the latter showed mainly elongated lamellae, that closely resemble those visible on cryo-EM images (Supplementary Fig. 17). Thickness, width and periodicity obtained from the corresponding AFM, SAXS and EM experiments closely match those recorded from the 24-mer MD simulations, which thus sheds light on the molecular level build-up of the observed supramolecules. Finally, to confirm the relevance of salt bridge formation between phosphates and the -NH$_3^+$ groups of the lysine residues, we have also tested this system at a pH higher than the pKa of lysines resulting in deprotonation (Supplementary Fig. 2c). Here the CD spectra confirmed that in this case the coassemblies could not form, consequently the presence of charged lysines is crucial for the lamellar assembly formation.

## LPS induces coassembly formation

To address whether phosphate-assisted coassembly formations could take place in the presence of bacterial membrane components, we studied lamellin-3K with vesicles composed of bacterial and mammalian lipids, and also extracellular vesicles having a complex bilayer composition. In particular, the main focus was on lipopolysaccharides (LPS), which is a complex macromolecule present in the cell wall of Gram-negative bacteria (Supplementary Fig. 21). Its lipid A part is rich in phosphate groups and it anchors LPS onto the outer membrane of the bacterial cell wall, whereas the anionic sugar moieties and the glycan in the core oligosaccharide are responsible for forming a permeability barrier that defends bacteria against chemical attacks[57]. The CD signal of lamellin-3K with LPS in pure water pointed toward the same conformational change as observed for the lamellin-3K in PBS (Fig. 2a). In addition, we could observe supramolecular patterns on TEM images which were similar to the striped lamellae observed earlier, though here the long LPS likely introduce more bends into the assemblies (Fig. 2c and Supplementary Fig. 22). The array widths observed for lamellin-3K in PBS and LPS were similar. Furthermore, the Lys-Lys distances observed in the MD simulations, indicate that lamellin-3K is highly able to adapt to the average interphosphate distances found for LPS ( ~1.2–1.5 nm)[23], and thus it is likely to be positioned near the lipid A region, forming an organized assembly [58,59]. To further analyze this interaction, we addressed the 3K-LPS system using IR spectroscopy. Based on spectral features of the peptide bonds, we can assume that lamellin-3K adopts regions with uniform conformation in the 3K-LPS complex (Supplementary Fig. 23b). Simultaneously, the bands corresponding to vibrations of the LPS phosphates become well-resolved only upon 3K addition (Fig. 2b), indicating a more ordered packing of phosphate moieties within the 3K-LPS coassembly. Importantly, lamellin-3K not only interacts with the phosphates from the lipid A region, but also with its acyl chain region. Perturbation reflected by the shift of ethylene vibrational bands indicates that lamellin-3K intercalates into the lipid chains[60] of LPS and thus the coassembly formation can continue through the hydrophobic lipid chains (Supplementary Fig. 23a). To address whether the 3K-LPS interaction is specific or similar coassemblies could also occur for 3K with other phosphate-containing membrane components, we also tested 3K with vesicles composed of DOPC, DOPC/DOPG (8:2, PC/PG), brain total lipid extracts, and also *E. coli* polar lipid extracts (Supplementary Fig. 24). In addition, extracellular vesicles derived from red blood cells were also tested as these provide a complex mammalian membrane environment (Supplementary Fig. 25). In contrast to the CD pattern obtained with PBS or LPS, none of the above liposome system resulted in similar spectra as lamellin-3K in PBS, suggesting that LPS has a rather specific interaction with 3K.

## Potent and selective antibacterial activity - lamellae incise the cell envelope

To address whether these supramolecules could be formed on bacteria and exert antimicrobial activity, we performed growth inhibition

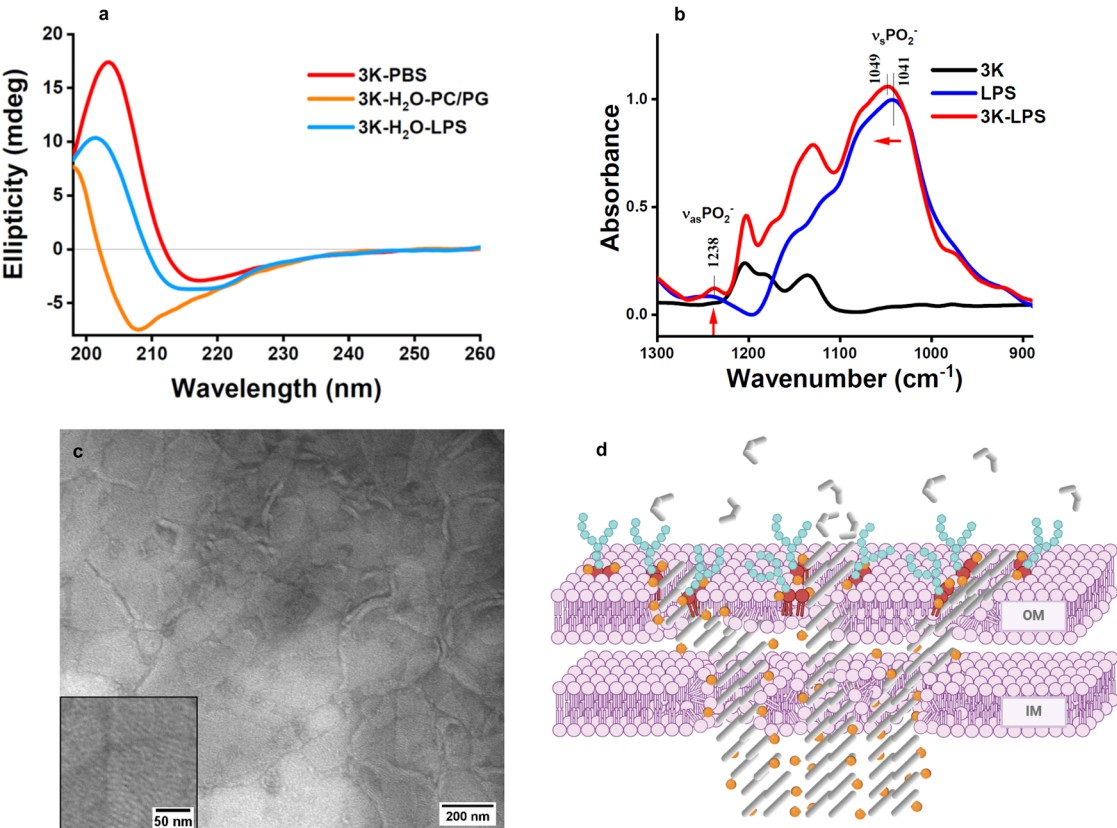

**Fig. 2 | Interaction of lamellin-3K with Lipopolysaccharides (LPS).**
**a** Representative circular dichroism spectra of 3K in PBS, in the presence of LPS and PC/PG liposomes displayed as red, blue and orange lines respectively (for more details, see Supplementary Fig. 2b, c). The intense maximum indicating ordered assemblies are present in PBS and with LPS but not with PC/PG (for vesicles with other lipid compositions, see Supplementary Figs. 24 and 25). **b** Phosphate vibrational region of IR spectra of 3K in water, LPS and 3K upon interaction with LPS represented by black, blue and red lines respectively. Well-resolved phosphate vibration bands are detected only for complexed LPS. **c** NS-TEM images of LPS treated with 3K showing curved striped supramolecular morphology. Scale bar: 200 nm (inset shows the lamellar morphology. Scale bar: 50 nm). Measurements

were repeated three times and similar results were obtained. **d** Schematic representation of the membrane disruption model by the in situ forming peptidic supramolecules. Interaction with LPS in the OM initiates peptide assembly as indicated by the curved, less regularly packed morphology and molecular build-up as shown in Fig. 1e. Elongation of the assembly beyond the cell wall towards the cytosol is assisted by intracellular phosphate ions resulting finally in long, well-structured lamellae protruding deeply into the bacterial cell. The monomeric and coassembled form of 3K is represented by curved and elongated gray lines respectively, the blue structures indicate the core and the O-antigen of the LPS, the red structures stand for lipid A with phosphates on either side represented by the orange spheres. Figure created with Biorender.com.

assays under the various conditions set above. Both compounds showed a strong antibacterial activity against *E. coli* in all tested media, with 3K being more potent than 2K (Fig. 3b). Lamellin-3K exhibited the highest antibacterial activity in phosphate-free medium, the decrease of viability starting well in the submicromolar concentration range with an $IC_{50} = 1.24\,\mu M$ (Supplementary Table 6). Upon addition of lamellin-3K to the bacteria, instantaneous bacterial viability decrease was observed (Supplementary Fig. 26). Excitingly, when we combined the phosphate-free conditions with negative stain-TEM (NS-TEM) and a high resolution setup to identify supramolecules, lamellae were detected in bacteria with a similar build-up as for lamellin-3K in PBS (Figs. 3a, 4a and Supplementary Fig. 27a–c). Cryo-EM images using the same bacterial setup showed a very similar scenario, i.e. membrane perturbation and disruption of the bacteriall cell membrane, with the local presence of lamellar morphologies (Fig. 4 and Supplementary Fig. 28). Coassembly of the designed supramolecules initiated close from the surface of the outer membrane (OM) and by penetration they spanned through the inner membrane (IM) deep into the cellular environment (Fig. 4a). At the position of the lamellae, the OM and IM boundaries became diffuse, suggesting that formed coassemblies incised the cell envelope, allowing leakage of the intracellular fluid (Figs. 3a, 4a and Supplementary Fig. 27d). To confirm the disruption of the OM, leakage assays were performed on bacteria. The results

indicate cell wall damage induced by lamellin-3K in a concentration dependent manner, with a maximum at ~20 μM 3K (Fig. 4c and Supplementary Fig. 29). Considering the phosphate-free media used for the recorded TEM images, it is probable that lamellin-3K coassembly formation starts with association to LPS on the OM, while fueled further mainly by the intracellular phosphate content. Interestingly, approximately 5 to 20 ordered supramolecular lamellae can be witnessed on the damaged bacteria (Fig. 3a). Furthermore, all the observed supramolecules exhibit comparable ordering and a relatively narrow width and length distribution, typically 5−7 sets of double lamellin-3K arrays, with an average width of 28.7 ± 9.4 nm and a length of 345 ± 29.5 nm (Supplementary Table 4). The phosphate-free environment ensures that the supramolecular coassembly formation is initiated in situ on the bacterial membrane. However, for considering future applications, we performed additional antimicrobial assays in a more physiological medium, as well as tested cytotoxicity of lamellin-3K on human cells. In PBS, lamellin-3K still exhibited a significant antibacterial efficacy, $IC_{50} = 11.6 \pm 1.5\,\mu M$, which represents a potent activity when compared to similar antimicrobial peptides. This indicates that even the preformed or partially assembled supramolecules are active antibacterial agents. In contrast, when cytotoxicity was measured (MonoMac 6 human monocytes), no significant toxic effect was observed up to 80 μM concentration (Supplementary Fig. 30 and

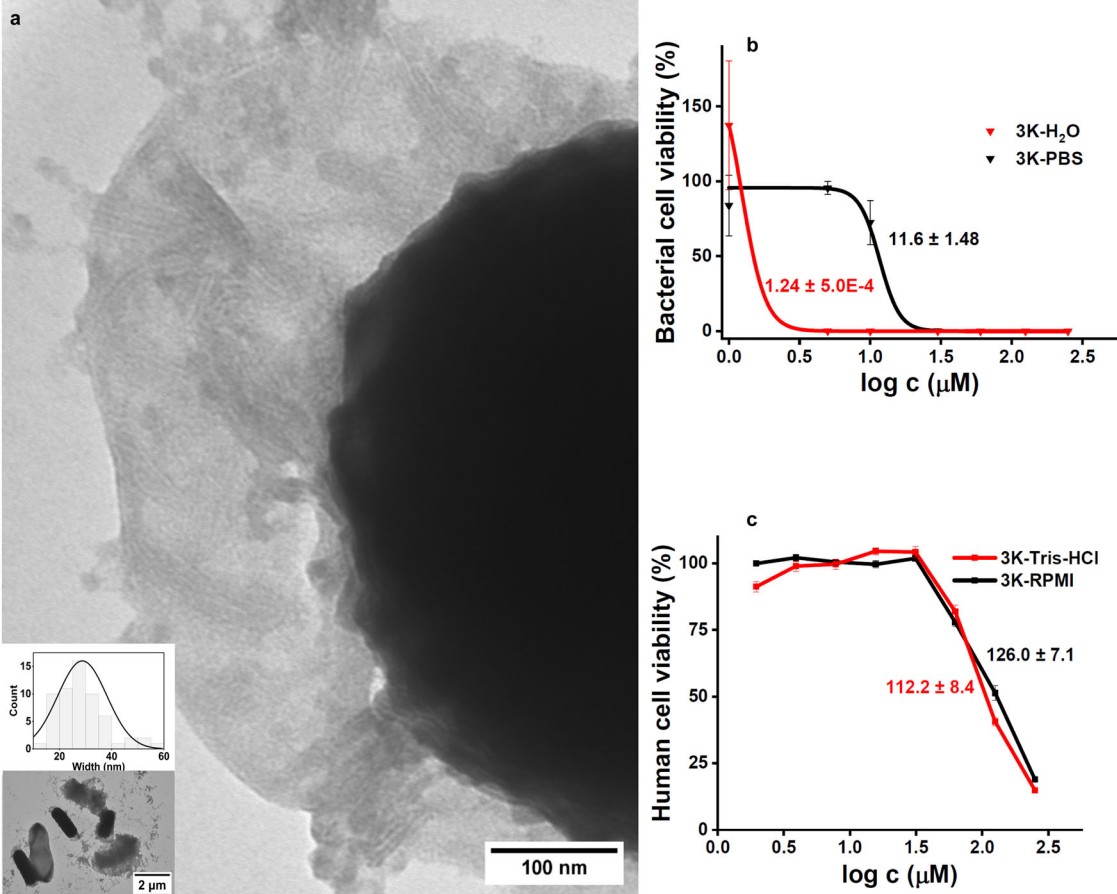

**Fig. 3 | Impact of lamellin-3K on cell wall integrity and viability. a** NS-TEM images of *E. coli* treated with 5 μM 3K in water, highlighting the supramolecular structures forming in situ. The leakage of cell content can also be clearly observed along with the diffused cell wall. Scale bar: 100 nm (inset shows the bacterial surroundings and displays the size distribution of lamellae width. Scale bar: 2 μm). For more details, see Supplementary Fig. 26 and Supplementary Table 6. Measurements were repeated three times and similar results were obtained. **b** Cytotoxicity activity of 3K against *E. coli* in water and PBS media represented by red and black lines (sigmoid fits), respectively. Note the higher antibacterial activity in water vs. PBS. The IC$_{50}$ values were calculated to be 1.24 ± 5.0x10$^{-4}$ μM for 3K in water and

11.6 ± 1.48 μM for 3K in PBS (mean ± SEM, *n* = 3 biological replicates). **c** Cytotoxic activity of 3K on human model cells (MonoMac 6) in Tris-HCl and RPMI media displayed as red and black lines respectively. No significant toxicity was detected up to 80 μM in phosphate-free and phosphate-containing media. The given values represent the calculated IC$_{50}$ values obtained from the dose-response curves, after fitting with non-linear regression using the GraphPad Prism program. The IC$_{50}$ values are represented as mean ± SEM which are 112.2 ± 8.4 μM for 3K in Tris-HCl and 126 ± 7.1 μM for 3K in RPMI media (*n* = 4, biological replicates). For more details, see Supplementary Fig. 30 and Supplementary Table 7.

Supplementary Table 7). The obtained IC$_{50}$ values were 126.0 ± 7.1 μM and 112.2 ± 8.4 μM, in a phosphate-containing (RPMI, 5.6 mM Na$_2$HPO$_4$) and in a phosphate-free medium (Tris-HCl), respectively (Fig. 3c). The surprisingly similar, low cytotoxic activity in these conditions ensures that lamellin-3K has low damaging effect on mammalian lipid bilayers both in its monomeric and in its preassembled form. Altogether, besides the cytotoxicity and antibacterial assays, the tests performed above on vesicles with various mammalian lipids, on vesicles with a number of bacterial lipids, and also on heparin-mimics containing sulfonates, it can be concluded that these overall results demonstrate a high selectivity of lamellin-3K towards the LPS-containing bacterial cell wall.

## Discussion
### Molecular details of the macroscopic morphology
The electron micrographs demonstrating antimicrobial supramolecule formation directly on a bacterial cell greatly promote our understanding on the mechanism of membrane targeting antibiotic assemblies. In combination with biophysical results, this insight shows that LPS in the OM is a key component that can trigger formation of the 3K-phosphate coassembly, resulting in several hundred nm long

lamellae. Interestingly, the segment of the coassembly which shows the formation of in situ supramolecular structures on the bacterial cell (Fig. 4 and Supplementary Fig. 28) resemble more to lamellin-3K with LPS, with curved stripes (Fig. 2c), but once it is beyond the OM and IM, it displays structured rectangular lamellar morphology resembling to those of lamellin-3K in PBS or in phosphate solution (Fig. 1d and Supplementary Fig. 13). We hypothesize that the initial LPS binding allows lamellin-3K oligomers to interact with the acyl chains of the lipid A region, which then provides access to the intracellular phosphate rich milieu of the cell (Fig. 2d). The MD simulations, the in vitro TEM and cryo-EM images, as well as those obtained upon the treatment of bacteria in vivo altogether allow a direct interpretation of the molecular morphology. The structural parameters of the 3K-phosphate coassembly are similar in all environments tested. The AFM results suggest that a single lamella has a height of ~3 nm, which corresponds to the ~3.1 nm length of a single layer of lamellin-3K molecules in zigzag conformation (Fig. 1e, f). EM images also support that one lamella consists of a single molecular layer, as several striped layers with different orientation can be identified in regions where lamellae are stacked on each other (Supplementary Figs. 14, 17). For further analysis, though both NS-TEM and cryo-EM images provide the same

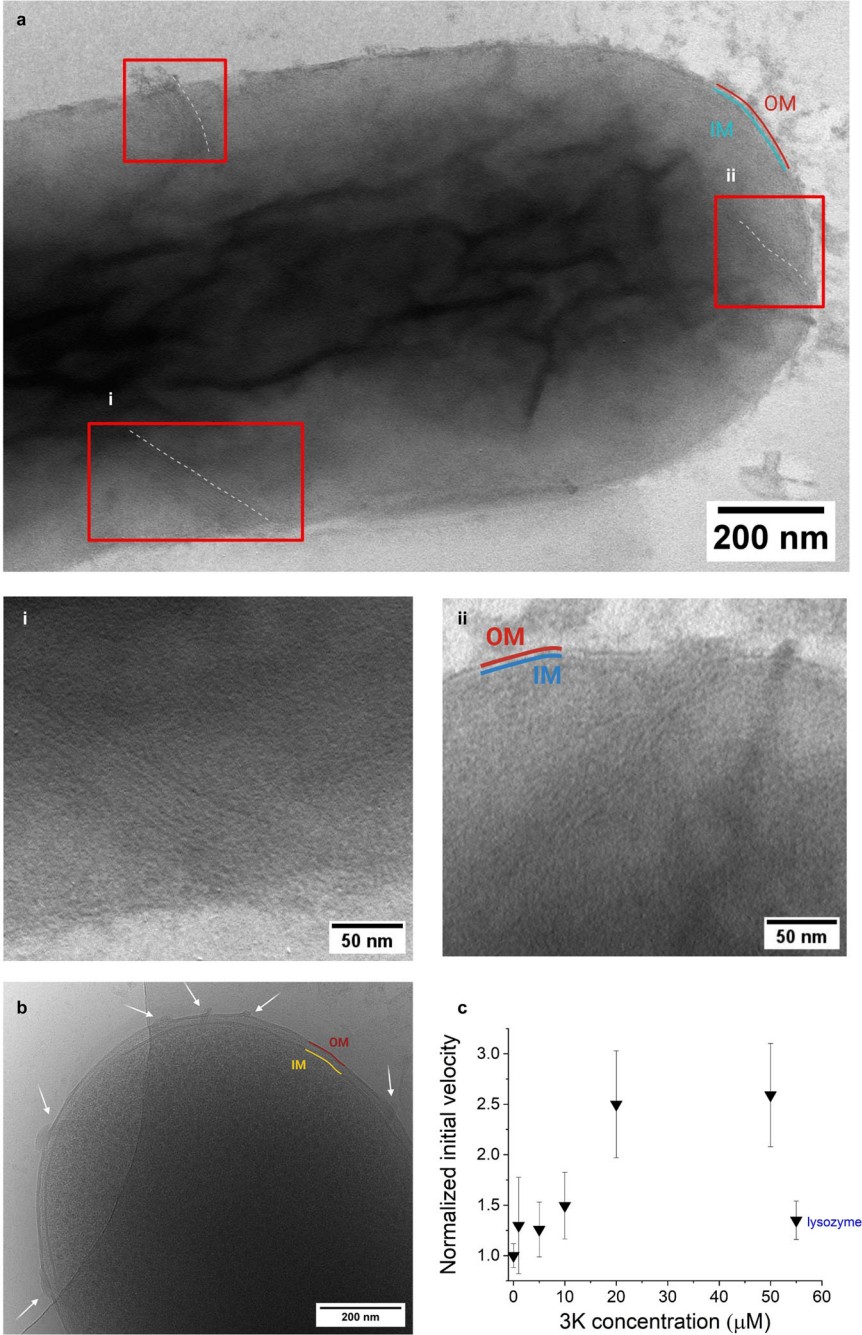

**Fig. 4 | Electron micrographs of *E. coli* cells treated with 3K show in situ formed supramolecular structures.** Bacterial cells were treated with 5 μM 3K in water for 120 min, and negative-stained with uranyl acetate. **a** NS-TEM image of a single bacterial cell highlighting the supramolecular structures formed within the cell. Scale bar: 200 nm (i–ii) The striped lamellae are observed protruding up to 100–200 nm deep into the cytoplasm from the bacterial cell wall. The inner and outer membranes (IM and OM) are also marked. Disrupted and diffuse OM and IM regions, and leakage of cell content can be clearly observed at positions where the lamellae enter the cell. Scale bar: 50 nm (for more details, see Supplementary Fig. 27 and Fig. 28). Measurements were repeated three times and similar results were obtained. **b** Cryo-EM image of a single bacterial cell with the supramolecular structures formed. Lamellar morphologies are highlighted by arrows. Scale bar: 200 nm. Measurements were repeated three times and similar results were obtained. **c** Normalized changes in outer membrane permeability induced by 3K. Normalized initial velocities of nitrocefin degradation following a 20 min incubation of bacterial cells with 3K in PBS at 1, 5, 10, 20, and 50 μM concentrations. Lysozyme was used as a positive control. Data points are the average of four biological replicates. Error bars indicate standard deviation. Normalization was done to the 0 μM nitrocefin concentration. For more details, see Supplementary Fig. 29.

qualitative information, the images where staining was applied provide improved visual insight, thus focus was put on those results. Interestingly, the lamellae in bacteria appear even more uniform than in vitro. Most of these supramolecules have six double arrays of lamellin-3K and their length also varies only to a small extent (Supplementary Table 4). This is surprising considering that they span nearly 350 nm in length on average. Such a morphology requires strong attractive electrostatic forces along the long axis. This is supported by the lack of supramolecule formation at a pH above pKa of lysines and by the MD models, where most lysines and phosphates have 3–4 coordinating salt bridges formed (Supplementary Figs. 2c, 31). However, the small variations in length and width suggest that

there may be additional intracellular species that control the coassembly formation. These striped lamellae are expected to form quickly in the presence of phosphates, which is supported by the immediate toxic effects observed visually during the bacterial assays. The acquired images demonstrate that the IM and OM are disrupted at the sites where the thin lamellae enter into the cells (Figs. 4a, 2d) and resulted in the leakage of the intracellular content, also confirmed by the leakage assay (Figs. 3a, 4c, Supplementary Figs. 27, 29).

### Efficient use of molecules for damaging the cell envelope

Considering that both lamellar width and length is well-controlled, and also that the 24-mer MD models give direct atomic details to 3K-phosphate coassembly, the obtained visual insight enables estimation of the number of molecules within the supramolecules penetrating the bacterial cell. Accordingly, ~10,300 ± 2500 lamellin-3K molecules are present in each lamella. Further on, we observe only ~5−20 lamellae in each of the cells that have major intracellular fluid leakage, thus we estimate that ~$5 \times 10^4$−$2 \times 10^5$ lamellin-3K molecules in the supramolecular form are sufficient to kill a single bacterium. For the same effect, similar natural membrane active compounds were estimated to require at least an order of magnitude higher number of molecules[61] rendering the current spontaneously forming coassembly highly efficient in exerting toxic activity. A word of caution, though the obtained data from the diverse experiments and simulations qualitatively support the above calculations, the solution phase information on morphology from the cryo-EM and AFM results suggest longer and thinner lamellae that may cause the above calculations less accurate. Further thorough studies with additional cellular assays and addressing e.g. equilibrium constants of the individual molecular forms are required for an improved quantitative insight into this direction.

The bacterial assays in PBS demonstrated that lamellin-3K maintains its potency even when preassembled, which suggests that upon interaction with a bacterial cell envelope, the preformed supramolecules can rearrange to penetrate the cell wall. This aspect is supported by the NMR measurements, which indicated that in equilibrium, monomers are also present (Supplementary Figs. 6, 7, Supplementary Tables 2, 3)[37]. Binding of lamellin-3K to the lipid A region of LPS may prove to be particularly beneficial against resistant species, as LPS-modifying enzymes in these bacteria leave the core of lipid A unmodified where several negatively charged phosphate moieties are present[62]. This phenomenon is expected to lower the probability of developing resistance against these species. The supramolecular morphologies presented here are so far unique for these non-natural compounds. This is also strengthened by the observation that the combined MD, NMR and TEM results ruled out the formation of helical structures, rather, these suggest a secondary structure closely resembling to extended sheets, that is atypical for acyclic β-peptides. From a more general point of view the lamellae obtained here show similarities to some of the oligomeric layers identified in simplified membrane models for natural peptide antibiotics[63]. The microscopic insight gained here using bacterial cells is expected to advance also our overall understanding on natural antimicrobial mechanisms. Owing to their macroscopic nature, the supramolecular structures can cause considerable damage to the bacterial cell membrane, which manifest a conceptually different mechanism compared to that of individual molecules. It is postulated that Nature regularly exploits such a technique and it may already be present for a long time in the arsenal of the host immune system akin to other supramolecular defensive agents[64].

## Methods

### Ethical Statement

The usage of human blood samples was approved by the Scientific and Research Ethics Committee of the Hungarian Medical Research Council (ETT TUKEB IV/701-3/2022/EKU) and during all investigations, we followed the guidelines and regulations of the declaration of Helsinki.

### Peptide synthesis

Solid-phase peptide synthesis of both peptides was carried out in a continuous flow reactor[65]. The coupling reagents and solvents used for the synthesis, such as 1,8-diazabicyclo[5.4.0]undec-7-ene (DBU), piperidine, trifluoroacetic acid (TFA), N,N-dimethylformamide (DMF), 1-[bis(dimethylamino)methylene]-1H-1,2,3-triazolo[4,5-b]-pyridinium-3-oxide hexafluorophosphate (HATU) and N,N-diisopropylethylamine (DIPEA) were purchased from Merck Life Science (Budapest, Hungary). TentaGel R RAM resin (0.19 mmol/g), purchased from Rapp Polymere GmbH, was loaded into the column (125 × 4 mm). Fmoc-protected amino acid (2.5 equiv), HATU (2.5 equiv) and DIPEA (5 equiv) were dissolved in 1.5 mL of DMF for the coupling. The flow rate was adjusted to 0.15 mL/min while the pressure and temperature were maintained at 60 bar and 70 °C respectively. 2% DBU and 2% piperidine in DMF were used for Fmoc-deprotection. After coupling and deprotection, the resin was washed with DMF. The peptides were then cleaved by stirring for 3 h in a solution of 95% TFA and 5% water. After evaporation of the TFA, the peptides were precipitated in cold diethyl ether. The crude peptides were purified by reversed-phase HPLC followed by lyophilization. The peptides were analyzed on an LC-40 HPLC system (Shimadzu, Kyoto, Japan) on a Phenomenex Jupiter Proteo C12 column (10 μm, 90 Å, 4.6 mm × 150 mm) using gradient elution with eluent A (0.1% TFA in H₂O) and eluent B (0.1% TFA in ACN: H₂O = 80:20 (v/v)). The flow rate was 1 mL/min, the gradient was 5−100 B% in 20 min (UV detection at λ = 214 nm). High-resolution mass spectrometry was performed on a Q Exactive Plus Hybrid Quadrupole-Orbitrap Mass Spectrometer (Thermo Scientific, Waltham, MA, USA). The yield after purification of 2K and 3K are 41% and 32% respectively. The purity of both 2K and 3K is greater than 95% as observed from the RP-HPLC chromatogram. Peptide mass: lamellin-2K, calculated for $[C_{46}H_{90}N_9O_7]$ ($[M+H]^+$) = 880.6963 Da, observed ($[M+H]^+$) = 880.6932 Da and lamellin-3K, m/z calculated for $[C_{60}H_{117}N_{12}O_9]$ M = 1,149.9066 Da, observed M = 1,149.9037 Da (Supplementary Fig. 32).

### Preparation of peptide solutions

The peptides were tested in pure MQ water, NaCl, phosphate solution and phosphate buffered saline (PBS). For preparation of the peptide solutions, 125 μM of 2K or 3K was used in different media (H₂O, pH 6.5, sonicated 10 min; NaCl 150 mM, pH 6.5; 10 mM phosphate solution (10 mM of disodium hydrogen phosphate, 2 mM of sodium dihydrogen phosphate), pH 7.4; and 10 mM PBS (137 mM NaCl, 2.7 mM KCl, 10 mM of disodium hydrogen phosphate, 2 mM of sodium dihydrogen phosphate), pH 7.4, sonicated 30 min) at 25 °C.

### Preparation of liposomes

**Preparation of model membranes.** High purity synthetic 1,2-dioleoyl-sn-glycero-3-phosphocholine (DOPC) and 1,2-dioleoyl-sn-glycero-3-[phospho-rac-(1-glycerol)], sodium salt (DOPG) were purchased from NOF (Tokyo, Japan). Liposomes were prepared by using the lipid thin film hydration technique. Lipids were dissolved in chloroform (LabScan, Budapest, Hungary) containing 50 vol% methanol (Reanal, Budapest, Hungary), which was then evaporated using a rotary evaporator. The resulting lipid film was kept in vacuum for at least 8 h to remove residual traces of solvent. The dried lipid film was hydrated with the assay buffer. After repeated heating (37 °C) and cooling (−196 °C) steps, the solutions were extruded through polycarbonate filters of 100 nm pore size (at least 11 times) using a LIPEX extruder (Northern Lipids Inc., Burnaby, Canada). The stock solution was prepared at 13 mM which was diluted 10 times for all measurements. For mimicking mammalian and bacterial cell membranes, pure DOPC and DOPC/DOPG (80/20 n/n%), were used throughout the study.

High purity *E. coli* Polar Lipid Extract and Brain Total Lipid Extract (porcine) were purchased from Avanti Polar Lipids Inc. (USA). Liposomes were prepared by using the lipid thin film hydration technique. Lipids were dissolved in chloroform (LabScan, Budapest, Hungary) containing 50 vol% methanol (Reanal, Budapest, Hungary), which was then evaporated using a rotary evaporator. The resulting lipid film was kept in vacuum for at least 8 h to remove residual traces of solvent. The dried lipid film was hydrated with PBS buffer and repeated vortex and sonication steps (each for 20 s) were performed to achieve a homogeneous mixture. The stock solutions were prepared at 10 mg/mL concentration and diluted 10 times for all measurements.

**Preparation of LPS**. The smooth (S)-form LPS from *Salmonella Minnesota* was kindly provided by Prof. Béla Kovács (Institute of Medical Microbiology and Immunology, University of Pecs, Hungary). LPS solution was prepared freshly prior measurement. Lyophilized LPS was suspended in deionized water (5 mg/mL) then 7 cycles of agitation (heating at 70 °C for 2 min, cooling to 5 °C for 2 min, followed by vortexing for 5 min) were applied. For 3K-LPS mixtures, a peptide-to-LPS mass ratio of 1:1 was maintained.

**Preparation of red blood cell-derived vesicles (REV)**. Healthy donors with informed consent were involved in our studies. Red blood cell-derived extracellular vesicle vesicles were used as a more complex cell membrane models and were isolated from 15 mL blood collected in tripotassium ethylenediaminetetraacetic acid (K3EDTA) tubes (Greiner). In the first step, the red blood cells (RBCs) were separated by centrifugation at $2500 \times g$ for 10 min at 4 °C (Nüve NF 800 R centrifuge) from plasma and buffy coat, and were washed with physiological salt solution at least three times at $2500 \times g$ for 10 min, 4 °C. Then RBCs were incubated in equal volume of phosphate buffered saline (PBS, pH 7.4) and stored for 7 days at 4 °C. After the end of the incubation period, the cells and the cellular debris were removed by differential centrifugation steps at $2500 \times g$ for 15 min and $3000 \times g$ for 30 min at room temperature. The supernatant was centrifuged at $16,000 \times g$ for 30 min at 4 °C (Eppendorf 5415 R, F45-24−11 rotor), after which the REV pellet was collected and re-suspended in 100 μL PBS. Prior the spectroscopic measurements, the REV samples were purified with size-exclusion chromatography (SEC) using a 3.5 mL Sepharose CL-2B gel (GE Healthcare, Sweden) gravity column. 100 μL of REV sample was passed through the column followed by the addition of PBS (900 μL) and the flow-through was discarded. The purified REV was eluted with 1 mL PBS and collected. The sample was stored at 4 °C until further use[66].

**Circular dichroism (CD) spectroscopy**
CD spectra were collected at room temperature using a JASCO J−1500 spectropolarimeter (JASCO, Tokyo, Japan). A rectangular quartz cuvette with a 0.1 cm path length (Hellma, Plainview, NY) was used in continuous mode between 190 nm to 260 nm with a scanning speed of 50 nm/min, a data pitch of 0.5 nm, response time of 4 s, bandwidth of 2 nm and 3 accumulations. 125 μM of the peptides (200 μL) were used with 150 mM NaCl, 10 mM phosphate and 10 mM PBS. For the pH-dependence studies, 125 μM of 3K-H₂O was prepared and the pH was adjusted to 12.7 using 10 M NaOH followed by the addition of 0.5 mM PBS. For the artificial model membranes, a concentration ratio of peptide-to-liposome of 1:10 was prepared and for the 3K-LPS mixtures, a peptide-to-LPS mass ratio of 1:1 was maintained. To visualize possible concentration dependent structural changes, the recorded CD data, ellipticity (Θ) in units of millidegrees (mdeg), were converted for the selected spectra (Supplementary Fig. 3) into molar circular dichroic absorption Δε, using the equation $\Delta\varepsilon = \Theta/(32982cl)$ where $c$ is the molar concentration of the peptide (in mol/L), and $l$ is the optical pathlength (in cm). All the spectra were corrected by subtracting a corresponding/matching blank.

**Attenuated total reflection-fourier transform infrared (ATR-FTIR) spectroscopy**
FTIR spectroscopic measurements were performed using a Varian 2000 FTIR Scimitar spectrometer equipped with a Golden Gate accessory (Varian Inc., Palo Alto, CA). 3 μL of the peptides in the aforementioned concentrations were pipetted onto the diamond ATR surface and the solvent was evaporated under ambient conditions to obtain a dry film. The spectra were collected at $2 \, \text{cm}^{-1}$ nominal resolution applying 64 scans. Data acquisition was followed by ATR correction, baseline correction and buffer subtraction. All spectral manipulations were performed using the GRAMS/32 software package (Galactic Inc, USA).

**Fluorescence spectroscopy**
Fluorescence spectra were recorded using a JASCO FP-8500 spectrofluorometer with an excitation and emission bandwidth of 10 nm and 20 nm, respectively. Three accumulations were measured each time. The 8-anilino-1-naphthalenesulfonic acid (ANS) fluorophore was excited at 388 nm and the emission was recorded from 410 to 600 nm. Binding assays were performed using 125 μM peptide (50 μL) and 2.5 μM ANS. Spectra were corrected for those of the corresponding peptide solutions.

**NMR Spectroscopy**
**Sample preparation.** For assignment purposes, appropriate amounts of lyophilized solid lamellin-2K (or lamellin-3K, respectively) were dissolved in H₂O to prepare ~1 mM solutions, followed by 15 min sonication and addition of 10% D₂O. For monitoring the effects of different solvents, the lyophilized lamellin-2K and lamellin-3K were dissolved in PBS, sonicated for 30 min, further on 10% D₂O was added. For 2K, the concentration of the lamellin was ~250 μM and the solution contained PBS. A sample containing 125 μM lamellin-3K in 90% H₂O/10% D₂O was titrated with a PBS stock (20 mM Na₂HPO₄, 4 mM NaH₂PO₄, 274 mM NaCl, 5.4 mM KCl) in 7 steps so that the concentration of Na₂HPO₄ was in the 125–1000 μM range. After each addition, the solution was sonicated for 30 min. All samples were measured at 298 K.

**NMR measurements.** All measurements were performed on a Bruker Avance III 700 MHz spectrometer (operating at 700.0 MHz for ¹H and 176.03 MHz for ¹³C) equipped with a Prodigy TCI H&F-C/N-D 5 mm probe head with *z*-gradient. Temperature was calibrated with 99.8% methanol-*d4*. Gradient calibration was done with the standard "doped water" sample. Resonance assignment and structural investigations were done using standard pulse sequences.

1D ¹H with excitation sculpting water suppression (number of scans was 64 for aqueous samples, 128 for 2K in PBS, and 1024 for 3K during PBS titration); ¹H-¹H TOCSY (MLEV17 spinlock with 80 ms mixing time, water suppression with 3-9-19 WATERGATE; for pure water samples: 800 complex points in the indirect dimension, 16 scans; for 2K in PBS: 512 complex points in the indirect dimension, 208 scans); ¹H-¹H ROESY (200 ms spinlock, water suppression with 3-9-19 WATERGATE; for pure water samples: 800 complex points in the indirect dimension, 16 scans; for 2K in PBS: 512 complex points in the indirect dimension, 240 scans); sensitivity-improved ¹H-¹³C HSQC spectra of lamellins in pure water (512 complex points in the indirect dimension, 32 scans) were recorded on natural isotopic abundance samples. ¹H and ¹³C chemical shift assignment was performed with NMRFAM-SPARKY[67] and data were deposited to BMRB with entry IDs 51854 and 51856 for lamellin-2K and lamellin-3K, respectively. For PFG-NMR experiments, the stimulated echo pulse sequence with bipolar gradient pulses was applied, using presaturation and the 3-9-19 WATERGATE sequence for solvent suppression. Linear incrementation of gradient strength was done in 16 steps between 5% and 95% of its maximal value. Gradient pulse lengths and diffusion delays were

optimized for each sample (Supplementary Table 2). Number of transients were adjusted to reach a sufficient signal-to noise ratio for each sample (at least 32 scans for ~1 mM sample in water and at least 256 scans for samples in PBS). Data evaluation was performed in TopSpin by fitting the Stejskal-Tanner equation on signals from the aliphatic region (0.7 − 1.8 ppm). PFG-NMR measurements were repeated at least three times and the resulted diffusion coefficients were averaged.

## Transmission electron microscopy

For analysing the solvent-dependant morphology and the interaction with LPS, 125 μM of the peptides were used. To analyse the formation of the in situ supramolecular structures, aliquots were collected from the 96-well microplates after the bacterial cells were treated with 5 μM 3K in water for 120 min. 2 μL of the samples were pipetted in each case to a 200-mesh copper grid (Ted Pella, Inc, California, USA) with a support film made of formvar. After a contact time of 1 min, excess liquid was removed, and samples were stained with 2% of uranyl acetate. Alternatively, the same procedure was followed for the staining agent, phosphotungstic acid (PTA) and the pH was adjusted to 7 using 10 M NaOH. TEM images were obtained routinely at magnifications of 11,000x, 28,000x, and 71,000x using a Morgagni 268D (FEI, The Netherlands) operating at 80 kV.

## Cryo-electron microscopy

For 3K-PBS, 5 μL of 125 μM lamellin-3K was applied to freshly plasma-cleaned TEM grids (Quantifoil, Cu, 300-mesh, R2/1) and vitrified into liquid ethane using Automatic Plunge Freezer EM GP2 from Leica Microsystems (8 °C, 100% rel. humidity, 300 s waiting time, 3.5 s blotting time). The grids were subsequently mounted into the Autogrid cartridges and loaded to Talos Arctica (ThermoScientific) transmission electron microscope for imaging. The microscope was operated at 200 kV. The micrographs were collected on Falcon3 direct electron detection camera at 92,000x nominal magnification with an underfocus in the range of 2−3 μm and an overall dose of ~40 e/Å$^2$.

For 2K-PBS and samples with *E. coli*, 4 μL of sample was applied to freshly plasma-cleaned TEM grids (Quantifoil, Cu, 300 or 200 mesh, R2/1) and vitrified into liquid ethane using Thermo-Scientific Vitrobot Mark IV (4 °C, 100% rel. humidity, 30 s waiting time (10 s for bacteria), 6 s blotting time (3 s for bacteria)). The grids were subsequently mounted into the Autogrid cartridges and loaded to Talos Arctica (ThermoScientific) transmission electron microscope for imaging. The microscope was operated at 200 kV. Cryo-TEM micrographs were collected on Ametek K2 direct electron detection camera at the 49,000x and 79,000x nominal magnification with the underfocus in the range 2−5 μm and the overall dose of 20 to 40 e/Å$^2$.

## Atomic force microscopy (AFM)

For AFM imaging in air, 1 μL droplets of serial dilutions of lamellin-3K (1,250, 125, and 12.5 μM, respectively) in PBS were deposited onto a cleaned Si (100) wafer chip and let to dry out by evaporation in ambient condition. AFM scans were performed at room temperature, using a Dimenson 3100 AFM instrument equipped with a NanoScope IIIa controller (Digital Instruments/Veeco, USA) in 512 × 512 pixel resolution. Scanning frequencies were either 0.2 or 0.5 Hz. Nano-sensors TM PPP-NCHR-20 type silicon cantilevers with the following parameters, mounted at 10° with respect to the sample stage plane, were used in Tapping Mode: thickness: 40 ± 1 μm; length: 125 ± 10 μm; width: 30 ± 7.5 μm; typical resonance frequency ~293 kHz; force constant: 10−130 N/m; aluminium-coated top; tip height: 10−15 μm; typical tip radius < 7 nm; tip half cone angle along the cantilever axis: 10°. To correct for tilt, respectively disjointed scanlines, a 1st order plane fit was applied to the raw image data,

followed by a 0th order flattening. For AFM imaging in liquid, a Nanosurf FlexAFM system (Nanosurf, Switzerland) was used operating in Tapping/Dynamic mode at room temperature. Budgetsensors Tap150GD-G type silicon cantilevers (Budgetsensors, Bulgaria) with reflective gold coatings with the following parameters were employed as probes: thickness: 2.1 ± 1 μm; length: 125 ± 10 μm; width: 25 ± 5 μm; typical resonance frequency in liquid ~47 kHz; force constant: 5 N/m (1.5−15 N/m); gold-coated top; tip height: 15−19 μm; typical tip radius <10 nm; tip half cone angle along the cantilever axis: 20°−25°. Freshly cleaved V-1 grade muscovite mica discs (NanoAnd-More Gmbh, Germany) were used as sample supports. 50 μL of 3K (125 μM) in PBS (500 μM) was deposited onto the mica disc and left for 5 min to adsorb onto the surface before starting the imaging. Topograpy data was recorded at 512 × 512 pixel resolution at randomly selected locations. The tip oscillation frequency in liquid was 47 kHz while the scan speed was 0.5 Hz.

## Small-angle X-ray Scattering (SAXS)

SAXS measurements on lamellin-3K (10 mM in water and 10 mM in PBS) were carried out using CREDO, our in-house pinhole SAXS camera[68,69]. Monochromatic Cu Kα X-rays (λ - 0.151 nm) are generated using a GeniX3D Cu ULD microfocus X-ray source (Xenocs SA, Sassenage, France). The sample was filled in a thin-walled borosilicate capillary of 0.01 mm nominal wall thickness and ca. 1.5 mm outer diameter, which was subsequently sealed using a hot glue gun. The capillaries were then placed into the temperature controlled (25 °C) sample holder block, situated in the vacuum chamber of the instrument. Scattering patterns were recorded with a Pilatus-300 k two-dimensional CMOS hybrid pixel detector (Dectris Ltd, Baden, Switzerland), located 411 mm downstream from the sample. Samples in water and in PBS were measured in two distinct runs. The basic data collection sequence started with measuring external (dark) and internal (empty beam) background images for 120 s each, followed by exposing a piece of 1 mm thick glassy carbon, used for controlling the beam intensity and for absolute intensity calibration[70,71]. Next, a capillary filled with a mixture of silver behenate and an SBA-15 mesoporous silica was exposed, also for 120 s, used for calibrating the angular range into momentum transfer units. After this calibration round, the sample was exposed for 300 s, repeated six times. The whole sequence was then repeated until the signal-to-noise ratio of the results was found satisfactory. Exposures of the same sample were collected, outliers were filtered and the remaining scattering patterns were averaged. The final patterns were azimuthally averaged, yielding the scattering curves, i.e., the scattered intensity vs. the momentum transfer (defined as $q = 4\pi \sin(\theta)/\lambda$ where $2\theta$ is the scattering angle). After eliminating exposures affected by spurious artifacts from background radiation, the total net exposure time amounted to 14 hours 20 minutes (172 exposures) and 18 h 45 min (225 exposures) for the sample in water and PBS buffer, respectively.

## Antimicrobial activity assay

To investigate the antimicrobial effect of lamellin-2K and lamellin-3K, we used the BL21 DE3 *Escherichia coli* strain with chromosomal resistance to chloramphenicol. The inoculum was grown overnight in the presence of 30 μM chloramphenicol. 0.5 mL of this inoculum was diluted into 50 mL fresh LB medium and grown till the optical density at 600 nm reached 0.5−0.6 indicating an exponential growth phase. Cells were then centrifuged at 3220 *g* at 4 °C for 20 min, washed twice in the assay medium (MQ water, 10 mM PBS, or 0.15 mM NaCl), then resuspended in 5 mL of the same medium. 10 μL cell suspension was diluted with the peptide solution to a final concentration of 250 μM, 125 μM, 60 μM, 30 μM, 10 μM, 5 μM, 1 μM and 0 μM (control) in 100 μl volume and incubated for 2 h in sterile, low-bind, U-bottom 96-well microplates (Greiner Bio-One, Hungary) with oxygen penetrating lid at

37 °C, at continuous shaking in a BioTek Synergy Mx plate reader. Aliquots from the wells were taken for the quantification of bacterial viability on LB agar plates after incubation at 37 °C for 16 h to obtain CFU/mL data. Treatments were performed in three biological replicates, each of which in three technical replicates.

### Leakage assay

Outer cell membrane permeability was measured using a nitrocefin-based leakage assay[72] executed as follows. BL21 DE3 cells (encoding chromosomal chloramphenicol resistance, and extrachromosomal penicillin resistance conveyed by beta lactamase on a pet15b plasmid), were grown in a liquid LB medium containing chloramphenicol (30 μM) and carbenicillin (130 μM) to early exponential phase (OD590 = 0.2–0.3). Cells were then centrifuged at 4 °C and 3220 $g$ for 20 min and washed twice with PBS. The cell pellet was re-suspended in 1/10 volume of PBS compared to the original culture volume. 10 μL of the cell suspension was diluted with the 3K peptide solutions to a final volume of 100 μL. Cells were incubated for 20 min in sterile, low-bind, U-bottom 96-well microplates (Greiner Bio-One, Hungary) with oxygen penetrating lid at 37 °C, at continuous shaking in a BioTek Synergy Mx plate reader 436. Following incubation of the cells with the peptide, nitrocefin (1 mg/mL solution freshly diluted with water from a 10 mg/mL DMSO stock solution) was added to a 50 μg/mL final concentration, and the chromogenic hydrolysis of nitrocefin was monitored by reading absorbance at 490 nm every 15 s for 100 cycles in a CLARIOstar plus plate reader (BMG Labtech). Initial velocities of the β-lactamase activity were evaluated from fits to the linear part of the progress curves.

### Cytotoxic activity

To evaluate the cytotoxicity of the compounds, AlamarBlue viability assay was performed on MonoMac 6 human monocytic cells (DSMZ no.: ACC 124, Braunschweig, Germany). MonoMac 6 cell line shows stable phenotypic and functional characteristics of mature blood monocytes therefore, it is a useful model for in vitro studies of monocyte biology and toxicity towards blood cells[73]. Cells were maintained as an adherent culture in RPMI-1640 media (Lonza, Basel), supplemented with 10% FCS, L-glutamine (2 mM) and penicillin/streptomycin (50 IU/mL and 50 μg/mL) at 37 °C in a humidified atmosphere containing 5% $CO_2$. On the day of the experiment, cells were washed twice with serum-free RPMI or with TRIS-HCl buffer (0.1 M, pH = 7.4) and plated in 96-well flat-bottom tissue culture plates (Sarstedt) (20,000 cells). 250 μM of the lamellin-3K was dissolved and serially diluted (125 μM, 62.5 μM, 31.25 μM, 15.63 μM, 7.81 μM, 3.91 μM, 1.95 μM) in the corresponding assay medium and added to the cells. The final concentration range was between 1.95 and 250 μM and the final volume was 100 μL. After 2 h of incubation, cells were washed twice and then 22 μL AlamarBlue (resazurin sodium salt in PBS, pH 7.4, c = 0.15 mg/mL) solution was added to each well. Following a 2 h of incubation, the fluorescence was measured at $\lambda_{Ex}$ = 530/30 and $\lambda_{Em}$ = 610/10 nm using a Synergy H4 multi-mode microplate reader (BioTek). The percent of viability was calculated compared to untreated control wells. All measurements were performed in quadruplets and the mean values together with ± SEM were represented. $IC_{50}$ values were calculated from the dose-response curves, after fitting with sigmoidal funcion using the GraphPad Prism software package.

### MD simulations

Molecular dynamics (MD) simulations were carried out using version 2021.6 of the GROMACS software[74], with our CHARMM force field extended for β-peptides[46]. The molecular models of lamellin-2K and lamellin-3K were prepared using PyMOL version 2.4, using the pmlbeta extension[75]. The lamellin-2K and lamellin-3K models were simulated in both monomeric and octameric forms both without phosphates and with phosphates (16). For lamellin-3K, a tetracosameric model was also built based on the octameric simulations

that contained phosphate ions (100). Phosphates in the CHARMM force field were modeled using methyl phosphate residues. Physiological conditions and system neutralization were achieved by setting NaCl to 150 mM in each simulation model. The temperature of the MD systems was set to 300 K in a 100 ps simulation run in the NVT ensemble using a Berendsen thermostat with 0.1 ps coupling constant[76]. Equilibration in the NpT ensemble followed using an isotropic Berendsen barostat set at 1 bar, with 2 ps coupling time. For the production runs, the velocity-rescaling thermostat with stochastic extension[77] and the Parrinello-Rahman barostat[78] were used. Long-range electrostatics were accounted for by the particle mesh Ewald algorithm implemented in the GROMACS program while the short-range Coulomb cut-off being 1.2 nm, dictated by the CHARMM force field. For additional information, please see Supplementary Information.

### Statistical analysis and reproducibility

All the data are expressed as mean ± standard deviation. The CD, IR, Fluorescence and NMR spectroscopy measurements were performed in triplicates for three independent sets of experiments. AFM, TEM and Cryo-EM were repeated three times, respectively and similar results were obtained. ImageJ software was used to calculate the length, width, array width and inter-array distance of the lamella for lamellin-3K in PBS, LPS and E. coli based on the TEM images. Antimicrobial activity assays were performed in three biological replicates, each of which in three technical replicates and cytotoxicity assays were performed in quadruplets and the mean values together with ± SEM were presented. For the leakage assays, data points are the average of four biological replicates.

### Reporting summary

Further information on research design is available in the Nature Portfolio Reporting Summary linked to this article.

## Data availability

The data that support this study are available from the corresponding authors upon request. The MD simulation files can be found at [https://doi.org/10.5281/zenodo.8363528] and the NMR chemical shifts are deposited to BMRB with entry numbers 51854 and 51856. A Source Data file containing cryo-EM micrographs, SAXS, TEM, NMR, and liquid AFM data is included with this manuscript. Source data are provided with this paper.

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

## Acknowledgements

This work was funded by the Momentum Program (LP2016-2 and LP2021-28) of the Hungarian Academy of Sciences, the National Competitiveness and Excellence Program (NVKP_16-1-2016-0007), the BIONANO_GINOP-2.3.2-15-2016-00017 project, and the National Research, Development and Innovation Office, Hungary (TKP2021-EGA-31, 2020-1.1.2-PIACI-KFI-2020-00021, 2019-2.1.11-TÉT-2019-00091, KKP_22 Project n.o. 144180, K131594 for J.M., K124900, K137940 for A.B., K142904 for Sz.B., and K138318 to J.T.). Support from Eötvös Loránd Research Network, Grant Nos. SA-87/2021 and KEP-5/2021, are also acknowledged. A.W. and Z.V. were supported by the János Bolyai Research Scholarship of the Hungarian Academy of Sciences. The authors acknowledge support from ELTE Thematic Excellence Programme 2020, the Szint+ Program, National Challenges Subprogramme-TKP2020-NKA-06. CIISB, Instruct-CZ Centre of Instruct-ERIC EU consortium, funded by MEYS CR infrastructure project LM2018127, LM2023042 and European Regional Development Fund-Project „UP CIISB" (No. CZ.02.1.01/0.0/0.0/18_046/0015974), is gratefully acknowledged for the financial support of the measurements at the CF Cryo-Electron Microscopy and Tomography. The authors thank Daniel Pinkas at CEITEC, Brno, for his excellent technical assistance and support. Fig. 1b (inset), Fig. 2d and Supplementary Fig. 21 have been created with BioRender.com.

## Author contributions

K.E.B., S.C., T.J. and T.B.-S. designed the study. A.W. performed the MD simulations. A.W. and Z.V. performed SAXS measurements. B.J. and Z.V. performed TEM measurements. K.E.B., T.J. and J.M. performed the ATR-FTIR measurements. K.E.B., M.Q.-P. and I.C.S. performed CD spectroscopy measurements. K.E.B. and T.J. performed the fluorescence measurements. C.L.S. and A.B. performed the NMR studies. L.R. performed the air-dried AFM measurements. G. Gy. performed the liquid AFM measurements. D.M. and J.T. performed and interpreted the antimicrobial activity and leakage assays. K.H. and S.B. performed the cytotoxicity activity. K.E.B. and I.M. performed the synthesis of the peptides. All the authors were involved in the evaluation of results. K.E.B., S.C., T.J. and T.B.-S. analyzed the overall results and wrote the paper. All the authors have reviewed and agreed to publish the final version of the manuscript.

## Funding

## Competing interests

The authors declare no competing interests.
