## [Peer Review File · Nature Communications]

In Situ Captured Antibacterial Action of Membrane-Incising Peptide LamellaeREVIEWER COMMENTS

Reviewer #1 (Remarks to the Author):

The manuscript by Beke-Somfai and coworkers describes a heterochiral, amphiphilic beta3-peptide able to self-assembly in lamellae. The work is of significance to the field, since to the best of my knowledge, there are no other examples in the literature of heterochiral, ACYCLIC beta3-peptides adopting sheet-like structures, although of course there are several good examples of heterochiral, alternating beta3-/beta2-peptides and heterochiral beta-peptides with CYCLIC residues. See for instance: T.A. Martinek, [https://doi.org/10.1002/1521-3773\(20020517\)41:10<1718::AID-ANIE1718>3.0.CO;2-2](https://doi.org/10.1002/1521-3773(20020517)41:10<1718::AID-ANIE1718>3.0.CO;2-2) and, especially: <https://doi.org/10.1021/ja063890c>; Seebach [https://doi.org/10.1002/\(SICI\)1521-3773\(19990601\)38:11<1595::AID-ANIE1595>3.0.CO;2-0](https://doi.org/10.1002/(SICI)1521-3773(19990601)38:11<1595::AID-ANIE1595>3.0.CO;2-0); and others.

In DOI: 10.1021/ol016868r, Sam Gellman and coworkers, after trying with acyclic beta3-residues (beta3-hLeu/beta3-hLys) to no avail, state (and they are not the only ones) that "cyclic residues (trans-2-ACHC) are important for self-association" and there are several works in the literature describing the need for cyclic or at least beta2,3-residues (with both C bearing side chains) to have sheet-like structures and self-assembly.

On the other hand, since the present manuscript is claiming a sheet-like structure on unprecedented beta-peptide systems, the data must unambiguously support such a claim. In the present version, the data presented by the authors are not convincing. There are several, critical issues, which unfortunately make the discussion not sound and the conclusions not supported, and hamper publication:

1) p.3, 'Structure and assembly in various environments'. As pointed out by the authors themselves, the CD profile of beta-peptides are difficult to unambiguously pair to specific 3D-structures. This is particularly true for 'random coil structures' and 'sheet-like structures' in beta-peptides, since their CD profile somehow overlap with other possible conformations. The claim that 'a minimum at ~ 213 nm, and a maximum at ~ 190 nm points to the predominance of a random coil structure' (legend to Supplementary Fig. 2) is not supported by a reference by the authors, since indeed it is not unambiguously assigned. The CD shape seems VERY similar to the 14-helix [Seebach: 'maximum at about 197 nm and a minimum at about 215 nm', in, among others: Seebach: *Helv.Chim. Acta* 79, 913-941 (1996) and *Helv.Chim.Acta* 79, 2043-2066 (1996)], which is a common 3D-structure for beta-peptides with acyclic beta3-residues like the ones herein described. How can the authors rule out the 14-helix conformation in phosphate-free environment? Besides, the 'sharp negative peak component developing at ~ 207 nm' has been identified as a 10/12 helix by Seebach (maximum at about 205 nm without an isodichroic point). Although the legend to supplementary figure 2 wisely states that this experimental observation supports a solvent-dependent conformational switch to a 'ordered structure' (by the way, perhaps not 'MORE ordered structure'), the main text (line 107) states: 'lamellin-3K adopted a more ordered structure similar to beta-sheet'. Perhaps that should read: 'the CD profile of lamellin-3K resembles that of sheet-like structures in beta-peptides'. In summary, supplementary figure 2a,b and ref. 30 do not give enough support to the claim of the peptide lamellin-3K adopting a sheet-like structure. A concentration-dependent study is needed. On the other hand, the conformational change in the presence of PBS is fully supported (also by supplementary figure 13), only it probably involves a switch from a 14-helix to a 10/12-helix. Also the ATR-IR results described (lines 108-110) do not support the conclusions drawn by the authors, and ref. 31 does not deal with beta-peptides at all (Andreas Barth, *Infrared spectroscopy of proteins*, BBA - Bioenerg. 1767, 2007, 1073-1101: Barth indeed talks about 'beta-(Ala)*n*', but it is actually protein Ala, the 'beta' being just the way they refer to beta-sheets at that time, while 'alpha-(Ala)*n*' referred to poly-Ala in alpha-helix. Indeed, Barth points to Krimm, S., & Bandekar, J. (1986). *Vibrational Spectroscopy and Conformation of Peptides, Polypeptides, and Proteins*. *Adv. Prot. Chem.*, 181-364. doi:10.1016/s0065-3233(08)60528-8, which in turn is based on the seminal work: Bamford et al., *Nature* 1953, 1954). There are FTIR studies on beta-peptides, such as Hetenyi et al, *J. AM. CHEM. SOC.* 2005, 127, 547-553, where indeed the spectra reported for helical beta-peptides are very similar to those reported in supplementary figure 3 of the present manuscript, but with very different

interpretation. The authors state (legend to supplementary Figure 3): 'The strong amide I band component centered at 1650 cm⁻¹ can be attributed to dominant random coil conformation while the shoulder at 1678 cm⁻¹ can be assigned to turn motives'. While in the abovementioned JACS2005, p 550, Martinek and Fülöp state: 'The high-frequency band (1679 cm⁻¹) was assigned to amide CO groups not involved as acceptors in H-bonding, and the low-frequency band (1648 cm⁻¹) to intramolecular H-bonds'. The change in band intensities is also attributed to switch between 14-helix and 12/10-helix. Thus a sheet-like structure is not proven by ATR-IR either.

The NMR in the presence of increasing PBS concentrations is reported. There are NMR studies of sheet-beta peptides in the literature, and the NMR signature to confirm that conformation is reported. There are several parameters that can be considered to identify sheet-like 3D-structure but none of them are reported by the authors. Supplementary information reports on the presence of unordered conformation in water, detected by 2D-NMR, although the reported portions of 1D and 2D spectra are not enough to support the claim. No comparisons with the many papers reporting NMR study on beta-peptides are drawn. The concentration of the two NMR studies seem to differ from each other. The solution is - probably - 1mM for the conformational analysis and surely 0.125mM for the self-assembly (legend to supplementary figure 5 does not report the concentration, but it seems that way from the experimental section).

The results of both NMR conformational analysis and DOSY (!) experiments are not mentioned in the main text: the authors seem to limit the NMR analysis to beta-peptide self-assembly, but 'microprecipitation' is not generally considered a NMR result. On the other hand, the same analysis in the presence of other negatively charged anions and at the same ionic strength (by salts other than phosphate) should be carried out, and far more data collected, before such a NMR-based observation of 'microprecipitation' be considered appropriate to support the authors' statement (lines 128-130): 'The NMR measurements thus support the charge driven assembly formation of lamellin-3K in PBS at the appropriate ratio of cationic and anionic residues, until a saturation concentration is reached'. By the way, where are the anionic amino acid residues?

The only support to the claim of the beta-peptide be in a sheet-like conformation is given by MD simulations, unfortunately not adequately described in the experimental section, and which seem to have been carried out on hydrogenphosphate ('divalent phosphate ions', legend to Figure 1) only. At least other divalent ions should be tested to claim for a probable selective effect due to phosphates. The DOSY results (not mentioned in the main text, but reported in the supporting material) do not seem to support the presence of an unordered structure in both aqueous and PBS (!) solution (the latter apparently going against the CD results) as claimed in the legend to Supplementary Table 1, and are fully compatible with the presence of 'helical' beta-peptides: they only rule out the presence of self-assemblies. In conclusion, there are no clear experimental evidence of peptide lamellin-3K adopting a sheet-like conformation in the presence of phosphate ions. The conclusions drawn by the authors in paragraph 'Structure and assembly in various environments', p.3, are not supported by the data. On the other hand, the self-assembly of HELICAL beta-peptides has been widely reported in the literature (see for instance Romila D. Gopalan, Mark P. Del Borgo, Adam I. Mechler, Patrick Perlmutter, Marie-Isabel Aguilar, Geometrically Precise Building Blocks: the Self-Assembly of β -Peptides, *Chemistry & Biology* 22, 2015, Pages 1417-1423, <https://doi.org/10.1016/j.chembiol.2015.10.005>). Although the presence of a sheet-like structure is not at all needed to obtain beta-peptide self-assembly in lamellae, its absence removes a lot of the work's significance and novelty.

2) The low-resolution ESI(probably) mass spectra reported by the authors in Supplementary Figure 19 do not provide enough characterization for the peptides. Yield of the synthesis, purity of the peptides from HPLC must be reported, and the HPLC chromatograms and High Resolution MS spectra enclosed as supporting material.

3) Supplementary Figure 6 seems to point to a better lamellae formation for 2K than 3K. Figure 6c does not seem to show lamellae for peptide 3K in 'phosphate solution', which is - probably -PBS, so this is a bit puzzling. Both figures 1d and supplementary figure 8 are not convincing evidence for 'striped lamella morphology' in PBS, especially when compared to the TEM image reported in Supplementary Figure 6e for 3K in PBS. SAXS also is not a direct proof of the onset of that particular type of self-assembly. The use of 'sheet-like' structure for the AFM images is incorrect, since it reminds of a 3D-conformation while here the authors probably just refer to the fibrillar assembly shown in

Supplementary Figure 10. Fibrillar morphology is quite common for beta-peptide self-assemblies. Both Figure 3 and Supplementary Figures 15 and 16 are also quite difficult to accept as a proof of lamellae formation in the presence of bacteria. The images have very different scale bars, but the lamellae seem the same in Figure 3b and Supplementary Figure 15a. Again, data do not support the claim of the authors for phosphate-driven lamellae formation for 3K.

4) The experimental data, including the estimated number of peptides per lamella, are not enough to support the statement (lines 41-43): 'EM micrographs on Gram-negative bacteria show that the lamellae incised the cell envelope causing growth inhibition already at SUBMICROMOLAR concentrations.' Besides, the antibacterial activity in water is difficult to assess, since bacterial survival can be granted only for a short time. This should be clearly stated and discussed. Therefore, it is risky to state that the IC₅₀ is smaller in water, since in that environment the bacteria are already suffering. The authors statement (legend to Fig. 3): 'Note the higher antibacterial activity in water vs. PBS' should be explained in view of this observation.

- Not enough details provided in the methods for the work to be reproduced. The current description of the MD methodology is not enough to reproduce the study. This holds true for other techniques too. NMR: number of scans is missing. TEM: '2 µL of the samples' is a bit too generic. Cryo-EM: peptide concentration apparently missing. The procedures followed for the antibacterial and cytotoxic assays should be better explained: number of bacteria (ORD 0.5-0.6) vs. number of cells (20,000)? Why 16h incubation for bacteria and just 2h for human cells? Peptide concentrations tested in the antibacterial assay are missing.

Minor points:

- Abstract. Line 32-33: please explain what is a 'mechanistic design strategy'.
- p. 3, line 102: 'Na₂HPO₄²⁻/NaH₂PO₄⁻' Please do change this (also in the supplementary material)
- p.3, lines 122-123: 'When varying the PBS concentration from 0 to 1000 µM,' more details on the concentration variations applied (on which of the PBS components) should be provided.
- p.4, lines 146-147: 'Lamellin-3K in PBS solution was also investigated by small angle X-ray scattering (SAXS).' Comparison with Lamellin-3K in pure water seems missing.
- p.5, line 210: Abbreviations should be explained (NS).
- p.6, lines 244-246: statement not clear, it should be rephrased.

Reviewer #2 (Remarks to the Author):

The work presented introduces heterochiral b-peptides rich in lysin, which they name lamellin-2K and lamellin-3K, as a promising alternative to current small-molecule antibiotics. These peptides are reported to form lamellar layers that cause inhibition of bacterial growth. Sometimes these lamellar peptides appear to penetrate the cellular membrane, causing leakage. The paper reads interesting.

Frequently, the paper lacks specific information, which is mandatory for the reader to judge the quality of the experiments and their interpretation. For example, the authors present many TEM images. It is not clear which are cryo-TEM images of vitrified samples and which are negative-stain TEM images of air-dried samples. Negative-stain TEM usually provides much higher contrast than cryo-TEM images. Thus, one must assume that the most main figures represent images of air-dried samples. Similarly, AFM images or SAXS measurements were taken of air-dried samples. The air drying of fragile samples such as most biological samples and peptides severely changes their structural state. Thus, one questions whether the structural measures taken are useful. A state of the art approach would be to take these measurements from a hydrated sample.

Another example is the statistics of the paper. In almost none of the figures the authors specify how many independently prepared samples were characterised, providing the same results. How representative are the images shown? Also, averages often are given as ranges instead of an average +/- deviation plus sample size.

The authors used DOPC and DOPC/DOPG to mimic mammalian and bacterial cell membranes. I do not

think these rather artificial lipid compositions can mimic mammalian and bacterial cell membranes at all. They represent a drastic oversimplification of cell membranes. Why did the authors not use at least *E. coli* polar lipid extracts, *E. coli* outer membrane vesicles, and mammalian membrane vesicles? At this point, I would advise you to do so.

Cytotoxicity. The authors describe the observation of the cytotoxicity of their compounds exposed to *E. coli*. However, at least a control could show whether this cytotoxicity also applies to living mammalian cells or not, instead of using a highly artificial lipid mixture to mimic the mammalian cell membrane.

Methods. Transmission electron microscopy. The authors write that the samples were pipetted onto the TEM grid. Which samples? At what concentration? In which solution?

Fig. 1d legend. Please specify whether this is a cryo TEM image or a negative stain TEM image. The high contrast suggests that this is an air-dried negative stained TEM image.

Figure 1h. Please explain what the EM image, including the inset, shows. What is the sample? Are these negative stain TEM or cryo-TEM images?

Figure 2c. What are 'NS TEM images'? Does NS stand for negative stain and air-dried liposomes? A striped morphology cannot be clearly resolved. The sample does not look like liposomes, rather than collapsed layers of membrane or lamellin. In addition, depending on the treatment, liposomes can show ripple phases, which is widely known. Show control images of LPS treated exactly the same way, but in the absence of 3K.

Figure 2c. How can the authors be sure that what they image is not only 3K, but 3K with LPS. Where is the control showing that both components have indeed mixed?

Figure 3. The *E. coli* imaged by TEM obviously was air dried. To unambiguously show that *E. coli* shows leakage, as claimed by the authors, so-called leakage assays should be applied.

All figures and tables. Please specify for every experimental result shown how many independent experimental repeats were done. for every average define the number of samples analysed and provide average and standard deviation or standard error.

Figure S4. Give the number of samples and independent repeats. provide averages and standard deviations for each bar shown.

AFM. Which imaging force has been applied to image the samples? As you may know the imaging force is crucial to image the accurate heights of peptide-based samples (besides imaging in solution).

Reviewer #1 (Remarks to the Author):

“The manuscript by Beke-Somfai and coworkers describes a heterochiral, amphiphilic beta3-peptide able to self-assemble in lamellae. The work is of significance to the field, since to the best of my knowledge, there are no other examples in the literature of heterochiral, ACYCLIC beta3-peptides adopting sheet-like structures, although of course there are several good examples of heterochiral, alternating beta3-/beta2-peptides and heterochiral beta-peptides with CYCLIC residues. See for instance: T.A. Martinek, [https://doi.org/10.1002/1521-3773\(20020517\)41:10<1718::AID-ANIE1718>3.0.CO;2-2](https://doi.org/10.1002/1521-3773(20020517)41:10<1718::AID-ANIE1718>3.0.CO;2-2) and, especially: <https://doi.org/10.1021/ja063890c>; Seebach [https://doi.org/10.1002/\(SICI\)1521-3773\(19990601\)38:11<1595::AID-ANIE1595>3.0.CO;2-0](https://doi.org/10.1002/(SICI)1521-3773(19990601)38:11<1595::AID-ANIE1595>3.0.CO;2-0); and others.

In DOI: 10.1021/ol016868r, Sam Gellman and coworkers, after trying with acyclic beta3-residues (beta3-hLeu/beta3-hLys) to no avail, state (and they are not the only ones) that “cyclic residues (trans-2-ACHC) are important for self-association” and there are several works in the literature describing the need for cyclic or at least beta2,3-residues (with both C bearing side chains) to have sheet-like structures and self-assembly. On the other hand, since the present manuscript is claiming a sheet-like structure on unprecedented beta-peptide systems, the data must unambiguously support such a claim. In the present version, the data presented by the authors are not convincing.

There are several, critical issues, which unfortunately make the discussion not sound and the conclusions not supported, and hamper publication:”

Q1: “p.3, 'Structure and assembly in various environments'. As pointed out by the authors themselves, the CD profile of beta-peptides are difficult to unambiguously pair to specific 3D-structures. This is particularly true for 'random coil structures' and 'sheet-like structures' in beta-peptides, since their CD profile somehow overlap with other possible conformations. The claim that 'a minimum at ~ 213 nm, and a maximum at ~ 190 nm points to the predominance of a random coil structure' (legend to Supplementary Fig. 2) is not supported by a reference by the authors, since indeed it is not unambiguously assigned. The CD shape seems VERY similar to the 14-helix [Seebach: 'maximum at about 197 nm and a minimum at about 215 nm', in, among others: Seebach: *Helv.Chim. Acta* 79, 913-941 (1996) and *Helv.Chim.Acta* 79, 2043-2066 (1996)], which is a common 3D-structure for beta-peptides with acyclic beta3-residues like the ones herein described. How can the authors rule out the 14-helix conformation in phosphate-free environment? Besides, the 'sharp negative peak component developing at ~ 207 nm' has been identified as a 10/12 helix by Seebach (maximum at about 205 nm without an isodichroic point). Although the legend to supplementary figure 2 wisely states that this experimental observation supports a solvent-dependent conformational switch to a 'ordered structure' (by the way, perhaps not 'MORE ordered structure'), the main text (line 107) states: 'lamellin-3K adopted a more ordered structure similar to beta-sheet'. Perhaps that should read: 'the CD profile of lamellin-3K resembles that of sheet-like structures in beta-peptides'. In summary, supplementary figure 2a,b and ref. 30 do not give enough support to the claim of the peptide lamellin-3K adopting a sheet-like structure. A concentration-dependent study is needed. On the other hand, the conformational change in the presence of PBS is fully supported (also by supplementary figure 13), only it probably involves a switch from a 14-helix to a 10/12-helix. “

A1: Thank you for these initial standpoints and the specific comments. We do share the enthusiasm toward β -peptide systems with Reviewer 1, however we also have to state that the most important finding of this work is that we developed thin lamellar antibacterial peptide assemblies which grow into bacteria from the cell wall (while no cytotoxicity) and that we were able to depict these by TEM *in situ* on bacteria. This is the reason why the manuscript did not focus in-depth on the potential secondary structures. Also, that was covered for very similar systems in our earlier paper, Szigyarto et al. *ChemSci* 2020 and we plan a dedicated follow-up study focusing exclusively on these peptide sequences.

Nevertheless, we do agree, that showing the nature of the lamellar formation is important also in this manuscript, and based on the above comments, we appreciate that those interested in beta-peptides will find additional structural info here relevant. Now we have extended the section 'Structure and assembly in various environments' (pages 2-3) and also added several related minor parts throughout the text. Furthermore, additional NMR, MD and CD/TEM experiments were performed as detailed below as response for specific questions of the Reviewer.

“How can the authors rule out the 14-helix conformation in phosphate-free environment?”

We made several additional simulations and experiments that clearly show that the alternating chirality (with always β^3 -position for the side chains), makes formation of helices very unfavourable.

1.) The MD simulation results on monomers both in Szigyarto et al. *ChemSci* 2020 and in the current submitted manuscript already rule out the presence of any helical structure in phosphate free environment. In addition, we have performed 6 new monomeric MD simulations with 3K restrained in helical conformations (H10, H12 and H14 as both right and left handed), in the presence of phosphates (Supplementary Figure 8). An additional

oligomeric MD was also run, where eight 3K were kept in H10 helical form (Supplementary Figure 9). The initial 500 ns restraints were applied for all 7 of these to allow for environmental optimization. Upon releasing restraints all the helical conformations become extended random conformations (or zig-zag with H-bonds for the octameric MD).

To aid clarity, we have inserted additional text in SI, and in the revised section '*Structure and assembly in various environments*' (page 3, lines 131-149). In addition, all simulation files are uploaded and available, please visit the link: https://zenodo.org/record/8388600?token=eyJhbGciOiJIUzUxMiIsImV4cCI6MTcyNTE0MTU5OSwiaWF0IjoxNjk2MjQwMDM0fQ.eyJkYXRhIjpw71nJlY2kljo4Mzg4NjAwfSwiaWQiOiM4MjllLCJyb29mQ0MjdlZiJ9.eNGJy3nOFWHniNoRq2NpX9wXAMGOMHq6hJTWI9mo_y5AI9AGRDB59F-qYMyQsD6pEtQX0cfdiVxg0hWdv0Ww for which now we granted open access to enable anonymous inspection. This will be available also for readers here.

2.) We have now extended details on the NMR measurements and in addition to pure aqueous solution, we executed a thorough 2D characterization of lamellin-2K in the presence of phosphates as well. NMR clearly shows lack of any defined helical secondary structures. Please see the additional answers related to NMR below, and also the details on NMR measurements '*Structure and assembly in various environments*' (pages 2-3, lines 108-126). Also, additional details and spectral regions are now inserted for NMR to the SI, please see Supplementary Figure 5,6 and Supplementary Table 1,2. Please note, that the NMR measurements also confirm that the 3K and 2K CD signals in water (3K-H₂O, 2K-H₂O) correspond to unordered, random coil structures.

3.) As a related request by Reviewer 1 in **Q4**, we made additional experiments with 2K and 3K and a heparin mimic, suramin a polysulfonated molecule that has a demonstrated strong helix inducing affinity as seen for more than 20 AMPs, for some examples see, Kohut *et al. Phys. Chem. Chem. Phys.* **2019**, *21*, 10644, Zsila *et al. RSC Adv.*, 2017, *7*, 41091-41097 and Quemé *et al. ChemBioChem* 2019, *20*, 1578). These show a distinct CD curve and a distinct morphology from that of 3K-PBS and 2K-PBS systems. This strongly suggests that the conformation of 3K and 2K in the lamellae are significantly different from these. These data on 3K/2K-suramin systems is now inserted to SI to Supplementary Figure 11 and detailed in the section '*Structure and assembly in various environments*' (pages 3, lines 150-156).

"Although the legend to supplementary figure 2 wisely states that this experimental observation supports a solvent-dependent conformational switch to a 'ordered structure' (by the way, perhaps not 'MORE ordered structure'), the main text (line 107) states: 'lamellin-3K adopted a more ordered structure similar to beta-sheet'. Perhaps that should read: 'the CD profile of lamellin-3K resembles that of sheet-like structures in beta-peptides'."

We agree that there was a confusion to some extent here. We have now rewritten the corresponding line to aid clarity. The comparison referred to the intensity rise seen upon formation of sheet structures of e.g. amyloids (page 2, lines 88-93). For Supplementary Fig.2 with think that 'more ordered structure' is a valid expression as this terminology clearly describes the increase of order.

"In summary, supplementary figure 2a,b and ref. 30 do not give enough support to the claim of the peptide lamellin-3K adopting a sheet-like structure. A concentration-dependent study is needed. On the other hand, the conformational change in the presence of PBS is fully supported (also by supplementary figure 13), only it probably involves a switch from a 14-helix to a 10/12-helix. "

We do agree with the reviewer that CD curves in general cannot be used to determine the absolute secondary structure of novel β -peptides. Also true, that general comparison with previous examples is further complicated as many times CD spectra were taken in methanol or DMSO, and those studies that contain both methanol and water phase CD spectra demonstrate several nanometer shifts between these solvents. Please note, that our main intention was to use the CD to identify, as a key quick reference, the formation of the lamellae, especially when using e.g. LPS. However, we have to agree that the discussions related to these should be changed as we immediately referred to sheets upon introducing the CD curves. Accordingly, we have now reworded the related sections in '*Structure and assembly in various environments*' (page 2, lines 88-93). However, based on the helical simulations and the additional experiments including NMR detailed above as answer for **Q1** we conclude that these β^3 -peptides with alternating chirality in the sequence do not form helices.

The repetitive LLKLLK (2K) and the LLKLLKLL (3K) are most likely arriving to similar secondary structures. Based on the striped lamellar morphology, these have the same molecular packing, yet, due to morphological differences (e.g. 2K and 3K lamellae have length differences) they arrive to different CD spectra. Due to the repetitive sequence and the lack of NMR crosspeaks, we would also rule out packing of turn motifs. To further support assemblies of extended conformation in sheet-like manner, we have also inserted more details on the IR spectra (see also for **Q2** below). These are very close to the ones in Szigyarto *et al. ChemSci* 2020 and the dominance of the 1680 cm⁻¹ band is also very similar to the one in the IR spectrum of sheet structures identified for cyclic β -peptides by Martinek *et al.* (Figure 4, *Angew. Chemie - Int. Ed.* **41**, 1718-1721 (2002)). Notably, the ratio of the 1680 cm⁻¹ and the 1650 cm⁻¹ intensities is clearly different for our lamellin-3K from those observed

for helical β -peptides Hetenyi et al. (*J. Am. Chem. Soc.* **127**, 547–553 (2005)). We now also inserted an extended discussion for the CD curves and inserted that the closest similar curve was found between the CD of 2K-PBS and a sheet-forming cyclic β -peptide determined by Martinek et al in ref. 33 (*Angew. Chemie - Int. Ed.* **41**, 1718–1721 (2002)).

According to the request of Reviewer 1, we have also performed several concentration dependent studies, which show the transition from one to another conformation. These are now in SI on Supplementary Figure 3 (shown below) and also mentioned in the above structural section.

Supplementary Figure 3: Concentration dependence of 3K. Concentration-dependent circular dichroism spectra of 3K in (a) H_2O , pH 6.5 at 25 $^\circ\text{C}$, (b) PBS 10 mM, pH 7.4 and (c) titration of 3K (125 μM) in water with 0-10 mM PBS. In water at 500, 250 and 125 μM concentrations, 3K show similar CD signature detected with a minimum at ~ 213 nm, which suggest the presence of the random coil structure. In contrast, in PBS (b), the formation of ordered assemblies can be detected with the gradual appearance of the maximum at ~ 207 nm. Additionally, to follow the formation of ordered assemblies in PBS (c), 3K in water was titrated with increased concentration of 2-10 mM PBS. Here, the transition from random coil structure at 0 μM of PBS towards more ordered assemblies can be trailed effectively.

Please also note, that the 2K and 3K NMR experiments in pure water clearly show the lack of helical structures, that correlates to our conclusions (Supplementary Figure 5,6 and Supplementary Table 1,2).

Please find our extended section, ‘Structure and assembly in various environments’, in the main body of the manuscript, and also the extended discussion related to this section in the Supplementary Information.

Q2: “Also the ATR-IR results described (lines 108-110) do not support the conclusions drawn by the authors, and ref. 31 does not deal with beta-peptides at all (Andreas Barth, *Infrared spectroscopy of proteins*, BBA - Bioenerg. 1767, 2007, 1073-1101: Barth indeed talks about 'beta-(Ala)n', but it is actually protein Ala, the 'beta' being just the way they refer to beta-sheets at that time, while 'alpha-(Ala)n' referred to poly-Ala in alpha-helix. Indeed, Barth points to Krimm, S., & Bandekar, J. (1986). *Vibrational Spectroscopy and Conformation of Peptides, Polypeptides, and Proteins*. *Adv. Prot. Chem.*, 181–364. doi:10.1016/s0065-3233(08)60528-8, which in turn is based on the seminal work: Bamford et al., *Nature* 1953, 1954). There are FTIR studies on beta-peptides, such as Hetenyi et al, *J. AM. CHEM. SOC.* 2005, 127, 547-553, where indeed the spectra reported for helical beta-peptides are very similar to those reported in supplementary figure 3 of the present manuscript, but with very different

interpretation. The authors state (legend to supplementary Figure 3): 'The strong amide I band component centered at 1650 cm^{-1} can be attributed to dominant random coil conformation while the shoulder at 1678 cm^{-1} can be assigned to turn motives'. While in the abovementioned JACS2005, p 550, Martinek and Fülöp state: 'The high-frequency band (1679 cm^{-1}) was assigned to amide CO groups not involved as acceptors in H-bonding, and the low-frequency band (1648 cm^{-1}) to intramolecular H-bonds'. The change in band intensities is also attributed to switch between 14-helix and 12/10-helix. Thus a sheet-like structure is not proven by ATR-IR either."

A2: We do agree that the text was confusing referring to both the α and β -peptide IR results. However, we do think that there are marked differences between the IR spectra obtained here, and the IR spectra obtained for helical structures. Furthermore, in the Martinek et al. (*Angew. Chemie - Int. Ed.* **41**, 1718–1721 (2002)) paper, the authors demonstrate how the IR spectra change, in particular the relative intensity of the 1680/1650 cm^{-1} bands when arriving to more mature sheet-like structures as the peptide sequence increases from trimers to heptamers. Our current IR spectrum shows a very similar gradual change from water to PBS, and the final results show high similarity to the IR spectrum of the β^3 - peptide with similar sequence in the Szigyarto et al. 2020 paper. Please note, that the dual presence of the 1680 cm^{-1} and the 1620 cm^{-1} shoulders suggest intermolecular H-bonding, and these shoulders can be identified in both studies with alternating chiral amino acid sequences. We have to agree though, that that IR is on dried samples, and IR sometimes has similar difficulties for comparison as was mentioned for CD spectra above.

We have extended the discussion on this which can be found in the main body of the manuscript under the section, 'Structure and assembly in various environments' (page 2, lines 93-107) and in Supplementary Fig.4, also shown below.

Supplementary Figure 4: ATR-IR spectra of 2K and 3K in different media report on peptide conformational arrangement in the peptide-phosphate coassembly. ATR-IR spectra of (a) 2K and (b) 3K were recorded for dry film samples obtained from solutions as described in Supplementary Figure 2. The IR spectra of 2K and 3K exhibit peculiar bands, namely amide I at 1600 - 1700 cm^{-1} and amide II at 1500 - 1600 cm^{-1} , characteristic for backbone amide bonds in peptides and proteins. For both peptides, the amide I band could be deconvoluted to three main components (see the representative examples for (c) 2K-water and (d) 2K-PBS). Based on band width, and the similarity to IR pattern of strand forming β -peptides and a peptide with close sequence build-up to 2K (peptide 5 in Szigyarto et al.¹⁴), the strongest amide I component centered at 1650 cm^{-1} could likely be assigned to an extended conformation with intramolecular H-bonds. Likewise, the shoulder at \sim 1680 cm^{-1} can be assigned to turn motives¹⁵ or backbone amide C=O groups not involved as acceptors in H-bonding.^{13,16} The low wavenumber band component at 1615-1620 cm^{-1} is indicative of intermolecular H-bonding. For 2K, the higher relative intensity of the shoulder at \sim 1680 cm^{-1} suggests a looser structure in non-phosphate media while the intensity gain of the component at 1615-1620 cm^{-1} is indicative of higher association of the peptide chains in

phosphate buffers. For 3K, the non-H-bonded amide C=O population is significantly reduced in the presence of phosphate ions, which is in line with more H-bonds formed in the phosphate-assisted coassemblies.

Q3: *“The NMR in the presence of increasing PBS concentrations is reported. There are NMR studies of sheet-beta peptides in the literature, and the NMR signature to confirm that conformation is reported. There are several parameters that can be considered to identify sheet-like 3D-structure but none of them are reported by the authors. Supplementary information reports on the presence of unordered conformation in water, detected by 2D-NMR, although the reported portions of 1D and 2D spectra are not enough to support the claim. No comparisons with the many papers reporting NMR study on beta-peptides are drawn. The concentration of the two NMR studies seem to differ from each other. The solution is - probably - 1mM for the conformational analysis and surely 0.125mM for the self-assembly (legend to supplementary figure 5 does not report the concentration, but it seems that way from the experimental section). The results of both NMR conformational analysis and DOSY (!) experiments are not mentioned in the main text: the authors seem to limit the NMR analysis to beta-peptide self-assembly, but 'microprecipitation' is not generally considered a NMR result. On the other hand, the same analysis in the presence of other negatively charged anions and at the same ionic strength (by salts other than phosphate) should be carried out, and far more data collected, before such a NMR-based observation of 'microprecipitation' be considered appropriate to support the authors' statement (lines 128-130): 'The NMR measurements thus support the charge driven assembly formation of lamellin-3K in PBS at the appropriate ratio of cationic and anionic residues, until a saturation concentration is reached'. By the way, where are the anionic amino acid residues?”*

A3: We agree with the reviewer about the lack of NMR spectroscopic data provided. Our updated discussion now includes a brief comparison with previous related NMR results as well. Indeed, the concentrations were adjusted as the Reviewer mentioned: 1 mM for conformational analysis in water (valid for both lamellins) and 0.125 mM for PBS titration of 3K. The reason of this difference is that NMR structural elucidation needed higher concentration (1 mM is sufficient), but this could not be achieved in PBS, as the solubility decreased drastically. PBS titration started with 0.125 mM 3K dissolved in pure water, this detail was added to the legend of Supplementary Figure 6. We further added the necessary experimental details to the Methods / NMR spectroscopy chapter (pages 7-8).

In the ‘Structure and assembly in various environments’ paragraph we include more results regarding the evaluation of the performed 1D and 2D homo-and heteronuclear NMR experiments (1D ^1H , DOSY, TOCSY, ROESY, ^1H - ^{13}C HSQC). All these modifications are highlighted with yellow (pages 2-3, lines 108-126). Also, the original supplementary figure of NMR data were separated into two figures, now they are labelled Supplementary Figure 5 and 6.

NMR characterization now is done for both lamellins in pure water and for 2K, in PBS as well. The following NMR data supported the presence of extended, disordered monomers in solution:

- low signal dispersion in both ^1H and ^{13}C , similar to Intrinsically Disordered Proteins (IDP)
- diffusion coefficients correspond to the D of an IDP with similar molar mass (about the evaluation of DOSY data, see our answers later)
- lack of long-range NOEs, only sequential connectivities in the ROESY spectra
- scalar coupling values of H^α - H^β and the small chemical shift difference of diastereotopic H^α resonances.

The figure below shows the full overlaid TOCSY (green) and ROESY (red-orange) spectra for 2K in pure water. The most important regions highlighted with black rectangles are magnified on Supplementary Figure 5,6. This figure clearly shows the low signal dispersion and the lack of long-range crosspeaks.

Based on these data, a helical structure can be excluded. On the other hand, $^3J_{\text{HN-H}\beta}$ coupling values were around 9 Hz. This observation suggests a somewhat restricted rotation around the N-C β bond, but this does not contradict our previous conclusions about an elongated conformation.

In PBS, the presence of phosphate greatly reduced the solubility of the lamellins. We detected no significant structural change based on 1D ^1H , DOSY and NOESY spectra, chemical shift perturbation was minor and nearly uniform for all ^1H resonances. Neither was any long-range NOE crosspeak detected. Also, DOSY showed no significant deviation of diffusion coefficients from those determined in pure water.

Q4: “The only support to the claim of the beta-peptide be in a sheet-like conformation is given by MD simulations, unfortunately not adequately described in the experimental section, and which seem to have been carried out on hydrogenphosphate (‘divalent phosphate ions’, legend to Figure 1) only. At least other divalent ions should be tested to claim for a probable selective effect due to phosphates. “

A4: We agree that descriptions were compact due to space considerations. Now we have provided more details, see section ‘Structure and Assembly in various environments’ (page 3, lines 131-149) also in the SI (Supplementary Figure 8-10). We have now also added details on the torsional angle parameters under the section, ‘Extended details on MD simulation results’ (pages 3-4, Supplementary Information). We now also direct the readers to our recently developed force field for β -peptides which is very capable of predicting the correct fold for numerous β -peptides tested (Wacha et al JCIM 2023 and Wacha 2019 CPC now Ref 45 and 46). In this recent paper we have tested several β -peptides with well-known secondary structures, and started simulations either from those determined (typically by NMR) or from a fully extended conformation. For all tested cases the simulations provided the correct fold with our force field, for example all those that were helical remained in that conformation (when started from their NMR structure), whereas when they were started from the extended one, these all folded spontaneously into their helical secondary structure. The force field was also able to correctly find turn-like and oligomerized structures. Since we use the same MD setup, in our opinion this puts high confidence in the MD results that we have performed for the current manuscript.

Note, that the new simulation mentioned above, where 8 pieces of 3K molecules were initially held in H10 helix conformation also demonstrate affinity toward the zig-zag structures and forming intermolecular H-bonds when restraints were removed. During the restrained part of the simulation H10 helices aligned into an aggregated octamer assembly. Once unrestrained, the helices unfolded very soon, out of the initial H-bonds in H10 conformation only a few remained by the end of the 500 ns fully relaxed run. This clearly demonstrates that neither the potential oligomerization, nor the presence of phosphates could stabilize helical secondary structures over the zig-zag conformation with interstrand H-bonds. This is now inserted in Supplementary Fig.8-10.

We have also considered the request of Reviewer 1 for divalent ions with experiments. Based on a coinciding request from Reviewer 2, we have now inserted tests on interactions of 3K with additional biological ions. Please note that in principle our focus is on biological environments and thus applying inorganic ions would shift the direction of the study to an unwanted aspect. Note also that we do agree with the intuitive feeling, namely that other ions with similar composition could trigger the same assembly formation as phosphate did here, also it is likely that other biological phosphates could trigger assembly as exemplified here by LPS as well. However, we think that it is the coordinating ability of anions, that often arising from the tetrameric arrangement e.g. for phosphates, is the key here for selective binding, as can also be observed for phosphates in e.g. enzyme active

sites. Thus, to consider this aspect in a biological setup, we have employed here a heparin-mimic compound, suramin, that is abundant with sulphonates and thus could represent biological macromolecules (heparin and heparin sulphates). This is also detailed as answers for **Q1** now. Note, that suramin is a very strong helix inducer for AMPs as we have noticed in several studies (*Phys. Chem. Chem. Phys.*, 2019, 21, 10644, *ChemBioChem* 2019, 20, 1578)). We have measured 2K and 3K with suramin both by CD and TEM. These results can be found in Supplementary Fig. 11. These indicate that although co-assembly formation is initiated by suramin with 3K, these morphologies are substantially different from the lamellar ones and also their CD spectrum is completely different. Thus, 3K interaction with a helix inducing molecule, with an adoptable backbone and 6 sulphonate anions, results in neither similar morphology, nor in similar CD spectrum (secondary structure) to that of 3K-PBS. Considering biological anion selectivity, with this addition and the others further detailed below for request of Reviewer 2, we compared 3K with phosphates (PBS and phosphate solution), with LPS, with NaCl, with DOPG, with a heparin mimic, with PS-rich red blood cell extracellular vesicles (REVs), with PE, PG and cardiolipin rich E.coli lipid extracts, and with PE, PS, PC, and PA containing brain lipid extracts. These results are presented in Supplementary Fig.22,23. Only PBS and LPS provided the characteristic CD pattern strongly suggesting that lamellin-3K selectively forms the described morphology with these.

Q5: “The DOSY results (not mentioned in the main text, but reported in the supporting material) do not seem to support the presence of an unordered structure in both aqueous and PBS (!) solution (the latter apparently going against the CD results) as claimed in the legend to Supplementary Table 1, and are fully compatible with the presence of 'helical' beta-peptides: they only rule out the presence of self-assemblies. “

A5: We agree with the Reviewer on the notion that insufficient details were provided on NMR, which we have now been addressed as detailed above. However, we have to disagree with the Reviewer regarding the DOSY results. In our previous works translational diffusion coefficients were evaluated based on the method reported by Dudás et al., (*Anal. Chem.* **91**, 4929–4933 (2019)), and Szabó, et al., (*Anal. Chem.* **94**, 7885–7891 (2022)) also supported by other approaches (Tang et al. *Phys. Chem. B* **31**, 5887–5895 (2022)). Hydrodynamic properties of folded proteins and intrinsically disordered proteins (IDPs) are distinct: a protein with a given molar mass possesses a larger diffusion coefficient when folded, as this state is more compact than the disordered state. Assuming that β^3 -peptides have similar effective densities as α -peptides and the same f correction factor of the Stokes-Einstein equation can be applied to both entities, we applied the empirical $\log M$ - $\log D$ formulas to the measured diffusion coefficients taking also into account the different experimental conditions (temperature, viscosity) (Szabó et al). The comparison of these corrected diffusion coefficients with the empirical formulas revealed that the lamellins are extended monomers in solution under both studied conditions, and the significantly more compact, helical structure can be excluded. However, differentiating a β -sheet tendency from a completely disordered state of the monomers is not possible with only the translational diffusion coefficient, as they possess very similar sizes, so this result does not contradict the conclusions derived from CD spectroscopy. The figure below shows the temperature-corrected D values of the lamellins fitted to the empirical $\log M$ - $\log D$ formulas.

“In conclusion, there are no clear experimental evidence of peptide lamellin-3K adopting a sheet-like conformation in the presence of phosphate ions. The conclusions drawn by the authors in paragraph ‘Structure and assembly in various environments’, p.3, are not supported by the data. On the other hand, the self-assembly of HELICAL beta-peptides has been widely reported in the literature (see for instance Romila D. Gopalan, Mark P. Del Borgo, Adam I. Mechler, Patrick Perlmutter, Marie-Isabel Aguilar, Geometrically Precise Building Blocks: the Self-Assembly of β -Peptides, Chemistry & Biology 22, 2015, Pages 1417-1423, <https://doi.org/10.1016/j.chembiol.2015.10.005>). Although the presence of a sheet-like structure is not at all needed to obtain beta-peptide self-assembly in lamellae, its absence removes a lot of the work’s significance and novelty.”

While we do agree with the Reviewer that the particular section needed to be restructured, we think that the novelty comes from the lamellar formation spontaneously on bacteria and the visual insight the imaging methods provide on this. While we are fully aware of the beautiful macroscopic structures formed from helical beta-peptides, it should also be mentioned that these materials are mostly much larger than ours, expanding well over 1 μm . In addition, we do not have a self-assembly phenomenon, but rather a phosphate-driven co-assembly, that clearly requires both lamellin-3K/2K and phosphate agents for formation. There is a very close agreement between the dimensions arising from the 24-mer MD values on the zig-zag sheet-like lamella and the TEM, AFM and SAXS values. Further on, the IR resemblance to previous sheet forming cyclic-peptides, the presence of intermolecular H-bonds based on the IR spectra, and the resemblance between the CD of 2K-PBS to again sheet-forming cyclic peptides we think all support the formation of sheet-like structures. On the other hand the NMR, the extended helical MD runs and the distinct pattern arising from a strong helix inducer all rule out the presence of helical beta-peptide structures.

Q6: *“The low-resolution ESI(probably) mass spectra reported by the authors in Supplementary Figure 19 do not provide enough characterization for the peptides. Yield of the synthesis, purity of the peptides from HPLC must be reported, and the HPLC chromatograms and High Resolution MS spectra enclosed as supporting material.”*

A6: Thank you for this observation. We have now included the RP-HPLC chromatogram and the HR-MS spectra in Supplementary Figure 29 of the revised manuscript. The yield after purification of 2K and 3K are 41% and 32% respectively. The purity of both 2K and 3K is greater than 95 % as observed from the RP-HPLC chromatogram. It has been included in the revised manuscript and highlighted in the ‘Peptide synthesis’ subsection of the Methods.

Q7: *“Supplementary Figure 6 seems to point to a better lamellae formation for 2K than 3K. Figure 6c does not seem to show lamellae for peptide 3K in ‘phosphate solution’, which is - probably -PBS, so this is a bit puzzling. Both figures 1d and supplementary figure 8 are not convincing evidence for ‘striped lamella morphology’ in PBS, especially when compared to the TEM image reported in Supplementary Figure 6e for 3K in PBS. SAXS also is not a direct proof of the onset of that particular type of self-assembly. The use of ‘sheet-like’ structure for the AFM images is incorrect, since it reminds of a 3D-conformation while here the authors probably just refer to the fibrillar assembly shown in Supplementary Figure 10. Fibrillar morphology is quite common for beta-peptide self-assemblies. Both Figure 3 and Supplementary Figures 15 and 16 are also quite difficult to accept as a proof of lamellae formation in the presence of bacteria. The images have very different scale bars, but the lamellae seem the same in Figure 3b and Supplementary Figure 15a. Again, data do not support the claim of the authors for phosphate-driven lamellae formation for 3K.”*

A7: We apologize for the confusion regarding the TEM images. We have amended this by including additional TEM images for 2K and 3K to clearly depict the observed morphologies. In the revised manuscript, Supplementary Figure 12a-c show the long, bundled, fibrillary structures obtained for 2K at lower and higher magnifications respectively while Supplementary Figure 13a-d depict shorter, rather edged, morphology observed for 3K. Also Figure 1d was altered to include additional insight by TEM.

Figure 3b represented a subsection of Supplementary Figure 15a wherein the lamella was observed to form inside the cell envelope of the bacteria. In the revised manuscript, we have now included Figure 4a, with the TEM image of a single bacterial cell highlighting the inner and outer membrane as well as the supramolecular structures formed within the cell in Figure 3a. The scale bars have been corrected and the regions where the lamella 3K assembly formation can be observed in sub-membrane regions are zoomed out for clarity. We now think that these newly inserted figures provide sufficient insight to confirm the lamellar morphology for 3K as well.

Regarding the AFM, Reviewer 1 has a valid point. The term “sheet-like structure”, in the caption of the SI Figure 10, referring to the case b), i.e. 125 μM , was used to describe the morphology of the objects measured, i.e. having one dimension considerably smaller than the other two (typical width 100-200 nm, typical length several 100 nm, while typical thickness only 6.4 nm). “Sheet-like” was meant as a synonym of ‘lamellar’, and does not refer to the beta sheet. The term “structure” was used in the sense of ‘object’ or ‘assembly’. This is now corrected in the revised Figure caption of Supplementary Fig.18.

Q8: “The experimental data, including the estimated number of peptides per lamella, are not enough to support the statement (lines 41-43): ‘EM micrographs on Gram-negative bacteria show that the lamellae incised the cell envelope causing growth inhibition already at SUBMICROMOLAR concentrations.’ Besides, the antibacterial activity in water is difficult to assess since bacterial survival can be granted only for a short time. This should be clearly stated and discussed. Therefore, it is risky to state that the IC50 is smaller in water, since in that environment the bacteria are already suffering. The authors statement (legend to Fig. 3): ‘Note the higher antibacterial activity in water vs. PBS’ should be explained in view of this observation.”

A8: A variety of methods can be used to evaluate the in vitro antimicrobial activity of a peptide. Considering that here we aim to provide specific information on the effect of the structural changes of the peptide in presence of key bacterial components (LPS), where phosphate components could bias the results, modified methods were needed using water as media. The Reviewer is probably right in that bacteria suffer in water. However, we compare the reduction of bacterial survival due to 3K to the already reduced survival rate without treatment as reference, thus with this we confirm that the shown effect is due to 3K, and therefore the observed inhibitory concentrations are as valid and justified for water as a test system as for other bacterial assay conditions. Since we were aware of these conditions, this is exactly why the same assays were performed in PBS as well. However, to strengthen this aspect as requested, please check lines page 5, lines 251-254.

IC20 determined shows indeed growth inhibition by 3K starting in the lower than micromolar concentration regime. We rephrased this sentence in the abstract and for clarity, we now include reference to Supplementary Table 4 (page 4, line 224) which shows this.

For additional mechanistic insight, as requested by Reviewer 2, leakage assays were also performed and included in Fig. 4b and Supplementary Fig.26. The results strengthen that the proposed mechanism takes place here, thus the opening of the cell envelope is supported by additional experiments. Regarding the estimated peptides per lamella, we feel that the additional MD simulations and the NMR/TEM experiments provide sufficient evidence for the claim of packing in an extended zig-zag conformation. Related to the request of Reviewer 2, additional phosphotungstic acid dyed TEM images were also taken to prove that the striped morphology was not a staining artifact (Supplementary Fig.13d). Using the alternative dye, striped morphology could still be observed. This result indicates that the stripes are due to the proposed co-assembly, and further strengthens our method for approximating the 3K content by arrays of peptide molecules. Based on all these observations, our calculations are straightforward even if they are only estimates.

Q9: “Not enough details provided in the methods for the work to be reproduced. The current description of the MD methodology is not enough to reproduce the study. This holds true for other techniques too. NMR: number of scans is missing. TEM: ‘2 μ L of the samples’ is a bit too generic. Cryo-EM: peptide concentration apparently missing. The procedures followed for the antibacterial and cytotoxic assays should be better explained: number of bacteria (ORD 0.5-0.6) vs. number of cells (20,000)? Why 16h incubation for bacteria and just 2h for human cells? Peptide concentrations tested in the antibacterial assay are missing.”

A9: We have added the necessary experimental details to the Methods of MD/NMR spectroscopy chapter as described above for **Q4** and **Q3**, respectively. We have now also included details on preparation of samples for TEM and the concentration of peptides used for the cryo-EM study in the respective sections. Peptide concentrations used for antibacterial assay has also been included. Additionally, we have also added a subsection on ‘Preparation of Peptide solutions’ under the Methods section (page 6, lines 337-342) for specifying the concentrations of peptide and solvents used throughout the study.

The number of cells in the bacterial and human cell cultures were set so that cells grow exponentially at the time of the treatment. In both cases, the incubation time with the peptide was 2h (120 min as originally written). The 16h incubation time refers to the overnight growth of the bacteria after streaking them onto plates to measure CFU (colony forming unit). CFU serves to quantify the effect of the applied treatment on bacterial cell viability. We rephrased the text to make these clear. Peptide concentrations tested in the antibacterial assay were shown in the figure showing “Bacterial cell viability” on a log scale (Fig.3b). For clarity, we now include them in the Methods section as well (page 9, line 501).

Relevant section of the Methods:

To investigate the antimicrobial effect of lamellin-2K and lamellin-3K, we used the BL21 DE3 *Escherichia coli* strain with chromosomal resistance to chloramphenicol. The inoculum was grown overnight in the presence of 30 μ M chloramphenicol. 0.5 mL of this inoculum was diluted into 50 mL fresh LB medium and grown till the optical density at 600 nm reached 0.5 - 0.6 indicating an exponential growth phase. Cells were then centrifuged at 3,220 g at 4 °C for 20 min, washed twice in the assay medium (MQ water, 10 mM PBS, or 0.15 mM NaCl), then resuspended in 5 mL of the same medium. 10 μ L cell suspension was diluted with the peptide solution to a final

concentration of 250 μM , 125 μM , 60 μM , 30 μM , 10 μM , 5 μM , 1 μM and 0 μM (control) in 100 μl volume and incubated for 2h in sterile, low-bind, U-bottom 96-well microplates (Greiner Bio-One, Hungary) with oxygen penetrating lid at 37 °C, at continuous shaking in a BioTek Synergy Mx plate reader. Aliquots from the wells were taken for the quantification of bacterial viability on LB agar plates after incubation at 37 °C for 16 h to obtain CFU/mL data. Treatments were performed in three biological replicates, each of which in three technical replicates.

Minor points:

Q10: “- *Abstract. Line 32-33: please explain what is a 'mechanistic design strategy'.*”

A10: We have now rewritten the abstract and changed the particular part for better readability.

Q11: “- *p. 3, line 102: 'Na₂HPO₄²⁻/NaH₂PO₄²⁻' Please do change this (also in the supplementary material)*”

A11: We have changed this in the revised manuscript and the Supplementary Information.

Q12: “- *p.3, lines 122-123: 'When varying the PBS concentration from 0 to 1000 μM ,' more details on the concentration variations applied (on which of the PBS components) should be provided.*”

A12: We have included the details with respect to specific concentrations used for the titrations in the ‘*Preparation of Peptide solutions*’ subsection as well as in the NMR spectroscopy section.

Q13: “- *p.4, lines 146-147: 'Lamellin-3K in PBS solution was also investigated by small angle X-ray scattering (SAXS).' Comparison with Lamellin-3K in pure water seems missing.*”

A13: We have included the SAXS of 3K in water in Supplementary Figure 17 of the revised manuscript. Also inserted the corresponding discussion to SAXS results and Methods.

Q14: “- *p.5, line 210: Abbreviations should be explained (NS).*”

A14: We have included the full form of the abbreviation NS in the revised manuscript.

Q15: “- *p.6, lines 244-246: statement not clear, it should be rephrased.*”

A15: We have rephrased this statement for clarity in the revised manuscript.

Reviewer #2 (Remarks to the Author):

“The work presented introduces heterochiral b-peptides rich in lysin, which they name lamellin-2K and lamellin-3K, as a promising alternative to current small-molecule antibiotics. These peptides are reported to form lamellar layers that cause inhibition of bacterial growth. Sometimes these lamellar peptides appear to penetrate the cellular membrane, causing leakage. The paper reads interesting. Frequently, the paper lacks specific information, which is mandatory for the reader to judge the quality of the experiments and their interpretation. “

Q1: *“For example, the authors present many TEM images. It is not clear which are cryo-TEM images of vitrified samples and which are negative-stain TEM images of air-dried samples. Negative-stain TEM usually provides much higher contrast than cryo-TEM images. Thus, one must assume that the most main figures represent images of air-dried samples. “*

A1: We do apologise for the confusion in this regard. We have now included this information in the legend of Fig.1d,2c,3a,4a and Supplementary Fig.11d,12,13,14,16,20,25 of the revised manuscript. Furthermore, to provide additional insight and meet requests raised with other comments, we have included additional EM images, where all figure captions clearly state their origin.

In addition to negative staining with uranyl acetate, we have performed staining with phosphotungstic acid as well (PTA). The TEM images taken confirm the formation of the same striped lamellar morphology for both staining complexes, confirming the formed morphology. These images are now inserted to SI to Supplementary Fig.13d and shown below. *Methods* section is also expanded by corresponding details. The discussion related to this is included in the section, ‘*Morphology and molecular packing*’ (page 3, lines 169-171).

Supplementary Figure 13: (d) NS-TEM images of 3K (125 μ M) in PBS stained with phosphotungstic acid. The same kind of lamellar morphology can be obtained with both staining agents for 3K in PBS.

Q2: *“Similarly, AFM images or SAXS measurements were taken of air-dried samples. The air drying of fragile samples such as most biological samples and peptides severely changes their structural state. Thus, one questions whether the structural measures taken are useful. A state of the art approach would be to take these measurements from a hydrated sample.”*

A2: We are sorry for non-clear description, the SAXS measurements were performed in solution phase, while the AFM was indeed recorded on dry samples. Considering altogether cryo-TEM, NS-TEM, AFM and SAXS, two of the methods describing morphology were used in solution phase/water-vitrified conditions, whereas two were employed on dried samples. However, considering the very close agreement between these (e.g. EM images), we think that the additional solution phase SAXS measurements we performed for 3K-water as answer for the specific request **Q13 for Reviewer 1** is sufficient to get a complete overview and the solution phase AFM would not add much more to the overall insight. Thus, we have now corrected the description for the SAXS measurements, added

the results on the additional SAXS experiments performed on the phosphate-free 3K, and also clarified that they were recorded in solution phase.

For this please see the Section ‘*Morphology and molecular packing*’ (page 4, lines 177-180) and Supplementary Figure 17 which is also pasted below for your reference.

Modifications in the manuscript:

3K both alone and in PBS solution was also investigated by small angle X-ray scattering (SAXS). The obtained curves indicated a regular periodic morphology, with a repeat distance of 2.86 nm using Bragg’s equation, which lies close to the EM values. The comparison of the solution phase 3K alone and 3K-PBS also demonstrated that the latter is in a more ordered state (Supplementary Fig.17).

Supplementary Figure 17: Periodic repeat distance determined from SAXS scattering pattern. The SAXS curve of 3K was measured at 10 mM 3K concentration in water and PBS. For PBS, two orders of equidistant peaks appeared, hinting at a regular periodic ordering of scattering units. The position of the first peak was determined by fitting to a Lorentzian function using an orthogonal distance regression (least squares) algorithm, yielding $2.197 \pm 0.005 \text{ nm}^{-1}$, which corresponds to a $2.860 \pm 0.006 \text{ nm}$ periodic repeat distance using the Bragg’s equation. In water, a similar peak is visible, at smaller q , corresponding to a larger periodic distance of $\sim 3.2 \text{ nm}$. However, no second harmonic is observable, suggesting that the periodicity has a shorter extent in space (i.e. not so long-range order) than found in the PBS solution.

Q3: “Another example is the statistics of the paper. In almost none of the figures the authors specify how many independently prepared samples were characterised, providing the same results. How representative are the images shown? Also, averages often are given as ranges instead of an average +/- deviation plus sample size.”

A3: Thank you for this comment. We have now improved clarity here by including a subsection on ‘*Statistical Analysis and Reproducibility*’ under the Methods section (page 10, lines 523-531) to address this. We have also included information with respect to this in individual figure captions as well as in all the sections pertaining to the characterization techniques in the revised manuscript. Further on, since we agree that this is particularly important for the biological measurements, the additional results performed for this revised manuscript, such as the newly inserted leakage assays, were also repeated to avoid biological variations, in this example we employed 4 biological replicates, each consisting of three technical replicates.

Modifications in the manuscript:

Statistical Analysis and Reproducibility

All the data are expressed as mean \pm standard deviation. The CD, IR, Fluorescence and NMR spectroscopy measurements were performed in triplicates for three independent sets of experiments. AFM, TEM and Cryo-EM were repeated three times, respectively and similar results were obtained. ImageJ software was used to calculate the length, width, array width and inter-array distance of the lamella for lamellin-3K in PBS, LPS and *E. Coli* based on the TEM images. Antimicrobial activity assays were performed in three biological replicates, each of which in three technical replicates and cytotoxicity assays were performed in quadruplets and the mean values

together with \pm SEM were presented. For the leakage assays, data points are the average of four biological replicates.

Q4: “The authors used DOPC and DOPC/DOPG to mimic mammalian and bacterial cell membranes. I do not think these rather artificial lipid compositions can mimic mammalian and bacterial cell membranes at all. They represent a drastic oversimplification of cell membranes. Why did the authors not use at least *E. coli* polar lipid extracts, *E. coli* outer membrane vesicles, and mammalian membrane vesicles? At this point, I would advise you to do so.”

A4: Simplified model membrane systems were used to demonstrate that the presence of these lipid compositions is not the sole prerequisite of assembly formation. Unfortunately, more complex lipid compositions and vesicles budding from the host membranes will also generate some simplifications compared to the original cell membrane composition and structure. Furthermore, the unknown components present in these renders such studies sub-optimal as deciphering results can potentially be challenging. This is especially the case for lipid extracts, e.g. Brain total lipid extract, where more than 50% of the total amount is unknown, where for e.g. the bacterial ones can be rather heterogenous and also result in lower concentrations that are not suitable for biophysical measurements.

However, we do agree that additional systems could still provide better insight or confirm result of the simple vesicles, thus, we have performed the same experiments for three additional systems as requested: liposomes made from *E. coli* polar lipid extracts, from brain total lipid extract, and on red blood cell-derived extracellular vesicles (REVs). In contrast, to the co-assemblies formation detected in the presence of LPS, none of the liposome systems (PC/PG, *E. coli* polar lipid extract and Brain total lipid extract) induce the formation of lamellin-3K assemblies in pure water (Supplementary Fig.22), Note that for REV experiment PBS buffer was used, as the pure water can contribute to the lysis of the vesicles. Moreover, the contribution of the large number of proteins present in the REVs (> 121, based on proteomic results) resulted in the appearance of a broad peak with a minimum \sim 225 nm (Supplementary Fig.23). In PBS buffer the intense peak at \sim 205 nm was preserved in the presence of PC and PC/PG liposomes and REVs. In contrast, in the presence of *E. coli* lipid extract and Brain lipid extract, the CD spectra lack the positive peak and resemble to the spectrum of 3K in pure water. The results detailed in the figures below indicate that only LPS generates similar CD spectra for 3K as in the 3K-PBS system.

The methods of preparations of liposomes made from *E. coli* polar lipid extracts, liposomes made from brain total lipid extract, and on mammalian red blood cell-derived extracellular vesicles (REVs) have been included in the *Methods* Section under the subsection, ‘*Preparation of liposomes*’ (page 7, lines 354-360 and lines 366-370). The explanations pertaining to these are now included in the text in the main body of the paper under the Section, ‘*LPS induces coassembly formation*’ (page 4, lines 212-218) as well as to the SI, Supplementary Fig.22,23.

Supplementary Figure 22: CD spectral pattern suggests minor peptide conformational changes upon binding to model vesicles. Formation of 3K-phosphate coassemblies was tested in more complex, but biologically relevant phosphate environments. Artificial model membranes such as liposomes prepared from i) the zwitterionic

DOPC, ii) DOPC/DOPG (8:2) iii) *E. coli* Polar Lipid Extract and iv) Brain Total Lipid Extract (from porcine brain) were employed. In water, only the interaction with the PC membrane showed a moderate change towards CD of 3K-PBS system, whereas all other tested systems showed clearly distinct CD curves from that. Here the vesicles from *E. coli* polar lipid extract indicate some formation of order, though the CD spectrum does not resemble to the 3K lamellar morphologies. In PBS, besides some intensity variation, the spectral pattern of 3K-PBS was preserved in the presence of PC and PC/PG liposomes. These results indicate that potential lipid binding does not affect markedly the solution-phase form of the peptides adopted prior interaction and that none of the environments induce formation of lamellar morphology akin to phosphates in PBS solution.

Supplementary Figure 23: CD spectral pattern of 3K in the presence of extracellular vesicles. Extracellular vesicles derived from human red blood cells (REVs) were used as a complex model membrane system to test its effect on co-assembly formation of 3K. In PBS, the preservation of the 3K-PBS morphology can be observed indicating that no significant interaction occurs between lamellin-3K and the REV vesicles. Note that for this experiment only PBS buffer was used, as in pure water strong REV lysis occurs. Moreover, the contribution of the large number of proteins present in the REVs (> 121, based on proteomic results¹⁷) can be attributed to the broad negative peak at ~ 225 nm.

Q5: “Cytotoxicity. The authors describe the observation of the cytotoxicity of their compounds exposed to *E. coli*. However, at least a control could show whether this cytotoxicity also applies to living mammalian cells or not, instead of using a highly artificial lipid mixture to mimic the mammalian cell membrane.”

A5: We think that our data might not be presented clearly for this particular case, as we have performed and inserted these mammalian tests. Cytotoxicity experiments are presented in Fig. 3c, Supplementary Fig.27 and Supplementary Table 5. All these were measured on a mammalian cell, namely MonoMac-6 human monocytes. To improve clarity, we have improved the figures and also made the description in the *Methods* section under the subsection, ‘Cytotoxic activity’ (page 9, lines 494-496), as well as in the discussion more straightforward (page 5, lines 251-254). These results clearly demonstrate that ‘cytotoxicity’ for the tested mammalian cells is nearly two orders of magnitude lower than for bacterial ones.

Q6: “Methods. Transmission electron microscopy. The authors write that the samples were pipetted onto the TEM grid. Which samples? At what concentration? In which solution?”

A6: This is again a valid point, we now have added the necessary experimental details to the TEM subsection under Methods. Additionally, we have also added a subsection on ‘Preparation of Peptide solutions’ under the Methods section for specifying the concentrations of peptide and solvents used throughout the study.

Please note, that since the issue of TEM, cryo-TEM images was raised in other comments, and also by Reviewer 1, we have inserted further TEM images to demonstrate the striped morphologies observed, as well as additional images to provide more insight to the bacterial systems and the nature of damage caused by 3K-lamellin. In addition, to exclude the effect of the staining complex, we have recorded TEM images using the stain phosphotungstic acid (PTA) as well (Supplementary Fig.13d). The discussion related to this is included in the section, ‘Morphology and molecular packing’ page 3, lines 169-171). The new TEM images are represented in Fig.1d,2c,3a,4a and Supplementary Fig.11d,12,13,14,16,20,25 of the revised manuscript.

Q7: “Fig. 1d legend. Please specify whether this is a cryo TEM image or a negative stain TEM image. The high contrast suggests that this is an air-dried negative stained TEM image.”

A7: The figure represents a NS-TEM image. We have included this in the figure caption of Fig.1d and also added an overview TEM image to enhance clarity.

Q8: “Figure 1h. Please explain what the EM image, including the inset, shows. What is the sample? Are these negative stain TEM or cryo-TEM images?”

A8: The figure 1h represented the NS-TEM image of the *in situ* formation of the supramolecular assembly of the 3K on a bacterial cell. In the revised manuscript, we represent this in a separate figure (Fig.4a) for better understanding. For further insight we have now inserted additional NS-TEM and cryo-TEM images to SI, please see Supplementary Fig. 11d,12,13,14,16,20,25.

Q9: “Figure 2c. What are ,NS TEM images ‘? Does NS stand for negative stain and air-dried liposomes? A striped morphology cannot be clearly resolved. The sample does not look like liposomes, rather than collapsed layers of membrane or lamellin. In addition, depending on the treatment, liposomes can show ripple phases, which is widely known. Show control images of LPS treated exactly the same way, but in the absence of 3K.”

A9: These are again valid points. NS stands for negative stain. We have included this in the figure captions of the revised manuscript. We have also added the NS-TEM image of the LPS control in Supplementary Fig.20a. The additional image clearly shows that there is a qualitative morphologic difference between the control and the LPS-3K lamellin system. Note that the sample has no vesicles, we studied direct LPS – 3K binding akin to others (Kaconis et al_Biophys J. 2011 100(11): 2652–2661, Li et al. Lipids and Lipoproteins, 2004, 279, 48, P50150-50156, Tang et al. Nat Comm. 2019, 10, 4175).

Q10: “Figure 2c. How can the authors be sure that what they image is not only 3K, but 3K with LPS. Where is the control showing that both components have indeed mixed?”

A10: This is closely connected to Q9. We have added the NS-TEM image of the LPS control in Supplementary Fig.20a for justifying our observations.

Q11: “Figure 3. The *E. coli* imaged by TEM obviously was air dried. To unambiguously show that *E. coli* shows leakage, as claimed by the authors, so-called leakage assays should be applied.”

A11: Thank you for this comment as this could clearly improve the study. Accordingly, we have performed a leakage assay and included these results into the revised manuscript. The assay performed shows that there is an increasing leakage with increasing 3K concentration that significantly deviates from the conditions when no 3K is present. This leakage assay was performed using 4 parallel measurements. After 3K concentration increases above 20 μ M, the extent of the leakage reached a constant level. This is much higher than for the positive control. It was a useful suggestion by the Reviewer as it confirms our previous mechanistic claim.

These results can now be found in Fig.4b and SI on Supplementary Fig.26 and in the main manuscript on page 5, lines 232-234 and page 5, lines 280-283. In addition, the description for the assays is now inserted to the *Methods* section (page 9, lines 478-491).

Q12: “All figures and tables. Please specify for every experimental result shown how many independent experimental repeats were done. for every average define the number of samples analysed and provide average and standard deviation or standard error.”

A12: We have now included a separate subsection on ‘Statistical Analysis and Reproducibility’ under the *Methods* section (page 10, lines 523-531) to address this. We have also included information with respect to this in the individual figure captions as well as in all the sections pertaining to the characterization techniques in the revised manuscript. The same procedure was applied to every newly inserted material where relevant.

Q13: “Figure S4. Give the number of samples and independent repeats. provide averages and standard deviations for each bar shown.”

A13: We have included this information in the caption of Supplementary Figure 7 of the revised manuscript.

Q14: “AFM. Which imaging force has been applied to image the samples? As you may know the imaging force is crucial to image the accurate heights of peptide-based samples (besides imaging in solution).”

A14: The force between the tip and the sample in conventional Tapping Mode AFM is much lower than the force in contact mode. Usually it is not measured routinely during the imaging, but its order of magnitude is 10 nN [Xu K, Sun W, Shao Y, Wei F, Zhang X, Wang W, Li P, *Nanotechnology Reviews* **7** (2018) 605–621; Kowalewski T, Legleiter J, *Journal of Applied Physics* **99** (2006) 06490]. Therefore, Tapping Mode AFM has been the primary choice over contact mode AFM for imaging soft, biological materials, such as amyloid fibrils [Adamcik J, Jung JM, Flakowski J, De Los Rios P, Dietler G, Mezzenga R, *Nature Nanotechnology* **5** (2010) 423–428.], DNA [Peters JP, Maher LJ, In: Dame R (ed.) *Bacterial Chromatin. Methods in Molecular Biology*, vol 1837, pp 211–256. Humana Press, New York, NY (2018)], self-assembled peptide bolaphiles [Zhao Y, Yang W, Wang D, Wang J, Li Z, Hu X, King S, Rogers S, Lu JR, Xu H, *Small* (2018) 1703216], etc. (With the advent of Bruker’s PeakForce Tapping mode, the tip–sample forces have been further reduced to the range of 10 pN.) In our AFM measurements, the tip–sample interaction, and the chance for any consequent damage or distortion of the sample was minimized by:

- 1) measuring dried (i.e. relatively stiff) samples in air, rather than hydrated (i.e., relatively soft) samples in liquid;
- 2) using Tapping Mode instead of contact mode;
- 3) measuring at frequencies very close (only 5% off) to the cantilever resonance frequency (the closer to the resonance frequency, the lower the force, see, e.g. [Kowalewski T, Legleiter J, *Journal of Applied Physics* **99** (2006) 06490];
- 4) keeping the amplitude setpoint of the cantilever as high as possible (close to the free oscillation amplitude).

This mode is regularly used for our peptide assemblies and so far no significant deviation from morphologies observed e.g. by TEM have been noticed. Note, that for soft matter samples where tubular regions would also appear such consideration need to be taken account with care as the apparent height can be distorted by even the very small forces applied.

REVIEWER COMMENTS

Reviewer #1 (Remarks to the Author):

The authors revised their work considerably.

Supplementary Figure 3: Concentration dependance(btw: dependence) of 3K." : CD spectra reported in this manuscript are not normalized for sample concentration. Although CD spectra should always be normalized for the concentration (i.e., not just ellipticity, but molar ellipticity), it can be accepted to report them as acquired. ON the other hand, since Supplementary Figure 3 reports a concentration-dependent study, it is of particular importance that at least the CD spectra in this figure are normalized for the concentration, otherwise non-specialist readers would get misled by the concentration-dependent increase of the signal intensity.

Abstract: "EM micrographs of Gram-negative bacteria show that the lamellae incised the cell envelope causing leakage with growth inhibition starting already at submicromolar concentrations". "EM micrographs" do not prove that the "growth inhibition starts already at submicromolar concentrations". The statement should read: "EM micrographs of Gram-negative bacteria show that the lamellae incised the cell envelope, while leakage and antibacterial activity assays prove that growth inhibition starts already at submicromolar concentrations". Indeed, in A8, the authors themselves state: "IC20 (IC50?) determined shows indeed growth inhibition by 3K starting in the lower than micromolar concentration regime."

Minor point: in the SI: please correct "dependance" to "dependence" also in the Table of Contents.

Reviewer #2 (Remarks to the Author):

I have now looked at the revised paper, which has improved considerably. However, I am not satisfied to read how the authors addressed some of my concerns, which I think are particularly important for a morphological study such as presented. For example:

1) In my first question I addressed my concern that TEM imaging of air dried samples causes artifacts and asked the authors to clarify whether their samples have been imaged in air or in the vitrified, native state. Apparently the majority of TEM images (Figs, 1, 2, 3, 4, S11, S12, S13, S14, S20) were imaged in air, which is not state of the art. Only one TEM image was taken of a vitrified specimen (Fig. S16). Its particularly not state of the art if the authors (as they do in their paper) characterize structural changes of air dried lipid bilayers. I think it would be better to address these structural changes in the as native as possible environment, which in this case would be the vitrified sample.

2) I further asked the authors to at least provide AFM images of hydrated samples. The authors declined since they do not think this would be useful. However, AFM imaging of air-dried samples in air is known to cause artifacts. For example the tubular structures made of lipids or peptide a prone to collapse. So the bilayer has been dehydrated by the authors and is not in its native structure anymore and the AFM imaging process in air is much more invasive compared to AFM imaging in fluid were forces can be better controlled and the sample remains in the native state. To get out of this dilemma the authors argue that in their paper they show that air drying causes no artifacts (which they don't actually because they merely compare air dried with airdried morphologies) and that AFM imaging of air-dried biological samples in air is state of the art and non-invasive (which is also not the case).

3) I further asked the authors to provide the number of samples analyzed. I thought the authors to provide these numbers explicitly for each experiment. Now there is a somewhat general statement in the Methods section, which is ok but not perfect. In addition the numbers of replicates are on the low

side. However, the authors should also show the data points for every graphs and provide average values and errors.

We would like to thank both Reviewers for the additional comments. Please find our point-by-point answers below the original comments.

Reviewer #1 (Remarks to the Author): The authors revised their work considerably.

Q1: *Supplementary Figure 3: Concentration dependence (btw: dependence) of 3K."*: CD spectra reported in this manuscript are not normalized for sample concentration. Although CD spectra should always be normalized for the concentration (i.e., not just ellipticity, but molar ellipticity), it can be accepted to report them as acquired. ON the other hand, since Supplementary Figure 3 reports a concentration-dependent study, it is of particular importance that at least the CD spectra in this figure are normalized for the concentration, otherwise non-specialist readers would get misled by the concentration-dependent increase of the signal intensity.

A1: This is a valid observation. We have now inserted graphs showing the relevant CD spectra in molar ellipticity as suggested into the revised version of the Supplementary Information (Supplementary Figure 3, panel a, and b). Please find these highlighted by yellow in SI. To avoid confusion, the original SI Fig. 3. is now split into two figures. All these changes are also noted in the manuscript wherever necessary. In addition, the related parts are modified in the methods section with details on how the graphs were obtained.

Q2: *Abstract: "EM micrographs of Gram-negative bacteria show that the lamellae incised the cell envelope causing leakage with growth inhibition starting already at submicromolar concentrations". "EM micrographs" do not prove that the "growth inhibition starts already at submicromolar concentrations". The statement should read: "EM micrographs of Gram-negative bacteria show that the lamellae incised the cell envelope, while leakage and antibacterial activity assays prove that growth inhibition starts already at submicromolar concentrations". Indeed, in A8, the authors themselves state: "IC20 (IC50?) determined shows indeed growth inhibition by 3K starting in the lower than micromolar concentration regime."*

A2: We have made the suggested changes in the abstract of the revised manuscript (page 1, lines 36-37). Please find it in yellow in these lines. We agree that this resulted a more accurate phrasing.

Q3: *Minor point: in the SI: please correct "dependance" to "dependence" also in the Table of Contents.*

A3: We have changed it accordingly in the Table of Contents as well as the figure caption of Supplementary Figure 3 in the revised version of the Supplementary Information.

Reviewer #2 (Remarks to the Author): I have now looked at the revised paper, which has improved considerably. However, I am not satisfied to read how the authors addressed some of my concerns, which I think are particularly important for a morphological study such as presented. For example:

Q1: *In my first question I addressed my concern that TEM imaging of air dried samples causes artifacts and asked the authors to clarify whether their samples have been imaged in air or in the vitrified, native state. Apparently the majority of TEM images (Figs, 1, 2, 3, 4, S11, S12, S13, S14, S20)) were imaged in air, which is not state of the art. Only one TEM image was taken of a vitrified specimen (Fig. S16). Its particularly not state of the art if the authors (as they do in their paper) characterize structural changes of air dried lipid bilayers. I think it would be better to address these structural changes in the as native as possible environment, which in this case would be the vitrified sample.*

A1: We might have misunderstood this question during the first revision. It is possible that the employed techniques could cause artifacts. Nevertheless, at this point allow us to state that we had no intent in circumventing crucial points and critical assessment of our results. On the contrary, we believe that we have performed a significant amount of experiments that are suitable for the various aspects of this study, and which were providing complementary insight on both solution phase (CD, NMR, fluorescence, SAXS) and dried samples.

Although we felt that our data supports our conclusions, we agree that potential artifacts should be excluded and minimised. We also agree that most images were from dried samples and since here morphology, especially on the bacteria, was a key point of the study, the images presented needed additional supporting micrographs to validate that they are not artifacts. Along this line, we have now acquired cryo-EM images of additional samples, in particular 3K-PBS was remeasured, and images on 2K-PBS were also recorded. In addition, we have recorded several cryo-EM images on bacterial samples with 3K in order to verify the morphologies in vitrified samples, closer to their native state. Please note, that staining gives better contrast for the images, thus those are preferred for demonstrating subtleties in these complex environments.

We have now inserted additional cryo-EM images to Figure 1, Figure 4, Supplementary Information (Fig. 13, Fig. 17., SFig 29, along with control images on E.coli. Fig. 29f). Please find all these in the revised version of the main body and in the revised Supplementary Information. In addition, we now adjusted the discussion accordingly in several places in the manuscript, all highlighted by yellow. Note, that the striped lamellar morphology is visible, it should also be mentioned that the lamellae are longer in the vitrified samples, as this was mentioned earlier too. Accordingly, some of the discussion is adjusted to consider this.

Please see these highlighted by yellow in the revised manuscript on lines 174-177, 188-192, 236-238, 271-272, 280-284 and 304-309. Please also see the corresponding measurement details in the Methods section under the subsection “Cryo-Electron Microscopy”.

Q2: *I further asked the authors to at least provide AFM images of hydrated samples. The*

authors declined since they do not think this would be useful. However, AFM imaging of air-dried samples in air is known to cause artifacts. For example the tubular structures made of lipids or peptide are prone to collapse. So the bilayer has been dehydrated by the authors and is not in its native structure anymore and the AFM imaging process in air is much more invasive compared to AFM imaging in fluid where forces can be better controlled and the sample remains in the native state. To get out of this dilemma the authors argue that in their paper they show that air drying causes no artifacts (which they don't actually because they merely compare air-dried with air-dried morphologies) and that AFM imaging of air-dried biological samples in air is state of the art and non-invasive (which is also not the case).

A2: As per the Reviewer's suggestion, we have now performed liquid AFM measurements and included these results on 3K in PBS in the revised version of the manuscript. The acquired images show lamellar morphologies, and are also in line with the cryo-EM images in that the observed lamellae are longer in solution phase. When comparing the air-dried and the solution phase AFM images, we conclude that at the same concentration (125 μ M), the two samples show the same height profile \sim 6.2-6.5 nm. In addition, several profiles recorded for the liquid AFM samples show a shoulder at \sim 2.6-2.8 nm, that corresponds to AFM profiles obtained for lower concentration, and suggest that we see double lamellae in the solution phase. Note, that liquid AFM measurements are particularly hard to obtain on these samples, since the phosphates likely neutralize the charged lysines during lamellae formation, and the formed supramolecules were less prone to attach to the bottom surface.

The additional images, along with an analysis on the height profiles of the found lamellae are all presented in the Supplementary Information. Please see Supplementary Fig. 20. The figure caption also contains height data collected from 50 measurements. In addition, an extended description is now inserted to the Atomic Force Microscopy subsection in the Methods. Furthermore extended discussions are inserted to the main body text, please find these on lines 188-191, and 304-309.

Q3: *I further asked the authors to provide the number of samples analyzed. I thought the authors to provide these numbers explicitly for each experiment. Now there is a somewhat general statement in the Methods section, which is ok but not perfect. In addition the numbers of replicates are on the low side. However, the authors should also show the data points for every graphs and provide average values and errors.*

A3: Please note, that besides the general statement in the Methods section, we have already included during the first revision details pertaining to statistics where we provided number of data points for graphs, and average values and errors at several places such as in figure captions and in the method descriptions of the particular techniques. However, we now try to further improve this aspect by including these details in all the figure captions and in the methods section of the manuscript. In addition, we have also mentioned at every instance the number of samples measured, where relevant or possible.

Related, we have provided expanded details for Small angle X-ray Scattering (SAXS).

All these are highlighted by yellow, please see the revised caption of Figure 3-4 and in the Methods section Small angle X-ray Scattering (SAXS), and in the Supplementary Information the following Tables and Figures:

In SI: Fig. 2., Table 1, Fig. 9, Fig 10, Table 3, Fig. 18, Fig 19, Fig 20, Fig. 26, Table 4, Fig. 29, Table 5.

Non-related to the comments, we have made a correction on the Figures of MD simulations in the Supplementary Information – values were erroneously displayed in nm, now corrected to Ångströms.

REVIEWERS' COMMENTS

Reviewer #1 (Remarks to the Author):

The authors addressed my concerns. The paper can be published.

Reviewer #2 (Remarks to the Author):

The authors represent a considerably revised manuscript in which they included state of the art cryo-TEM imaging and AFM imaging in the aqueous solution to characterize the samples. I have only two minor comments which the authors should address.

Please state how many independent simulations were taken for analysis of Suppl Figure 1. Please also state how many independent repetitions were done revealing same or similar results.

In Suppl Figure 8 please show all data points used to calculate mean and error bars.

We would like to thank both Reviewers for the additional comments. Please find our point-by-point answers below the original comments.

Reviewer #2

Q1: *Please state how many independent simulations were taken for analysis of Suppl Figure 1. Please also state how many independent repetitions were done revealing same or similar results.*

A1: We have now inserted this information accordingly to the caption of Supplementary Fig. 1 and also to the SI, in the subsection: Extended Methods for MD Simulations.

Q2: *In Suppl Figure 8 please show all data points used to calculate mean and error bars.*

A2: The table requested is now inserted to the source data file

	ANS fluorescence (arb. units)			Mean	Standard Deviation
	1	2	3		
2K-H ₂ O	1.49	1.52	1.56	1.52	0.04
3K-H ₂ O	3.86	3.10	2.66	3.21	0.61
2K-NaCl	1.25	1.26	1.21	1.24	0.03
3K-NaCl	7.84	7.97	7.54	7.78	0.22
2K-phosphate	11.79	12.30	10.90	11.66	0.70
3K-phosphate	21.15	19.38	17.29	19.27	1.94
2K-PBS	5.18	5.37	4.89	5.15	0.24
3K-PBS	14.50	14.90	13.42	14.27	0.77